# ON THE ROLE OF DISCRETE TOKENIZATION IN VISUAL REPRESENTATION LEARNING

**Tianqi Du**[1]*    **Yifei Wang**[2]*    **Yisen Wang**[1,3]†

[1] National Key Lab of General Artificial Intelligence,
   School of Intelligence Science and Technology, Peking University
[2] School of Mathematical Sciences, Peking University
[3] Institute for Artificial Intelligence, Peking University

## ABSTRACT

In the realm of self-supervised learning (SSL), masked image modeling (MIM) has gained popularity alongside contrastive learning methods. MIM involves reconstructing masked regions of input images using their unmasked portions. A notable subset of MIM methodologies employs discrete tokens as the reconstruction target, but the theoretical underpinnings of this choice remain underexplored. In this paper, we explore the role of these discrete tokens, aiming to unravel their benefits and limitations. Building upon the connection between MIM and contrastive learning, we provide a comprehensive theoretical understanding on how discrete tokenization affects the model's generalization capabilities. Furthermore, we propose a novel metric named TCAS, which is specifically designed to assess the effectiveness of discrete tokens within the MIM framework. Inspired by this metric, we contribute an innovative tokenizer design and propose a corresponding MIM method named ClusterMIM. It demonstrates superior performance on a variety of benchmark datasets and ViT backbones. Code is available at `https://github.com/PKU-ML/ClusterMIM`.

## 1 INTRODUCTION

Self-Supervised Learning (SSL) has recently emerged as a promising paradigm for learning meaningful data representations without access to labels. Besides the popular contrastive learning methods (Chen et al., 2020; Chen & He, 2020; Grill et al., 2020; Zhuo et al., 2023; Wang et al., 2021; 2023; Cui et al., 2023; Zhang et al., 2023a;b), there is a growing interest in masked image modeling (MIM) techniques. MIM requires masking parts of input images and subsequently attempting their reconstruction using the remaining unmasked parts. Notable examples like MAE (He et al., 2022), BEiT (Bao et al., 2022), PeCo (Dong et al., 2023), MaskFeat (Wei et al., 2022), and MAGE (Li et al., 2023) have demonstrated state-of-the-art performance on various downstream tasks.

Within the MIM domain, various reconstruction targets exist. For instance, MAE (He et al., 2022) employs the raw pixel values of the unmasked parts as its reconstruction target and MaskFeat (Wei et al., 2022) leverages features from other pretrained models as its reconstruction target, whereas some alternative methods adopt visual tokenizers to generate the reconstruction target (Bao et al., 2022; Dong et al., 2023). Such tokenizers convert image patches into predefined discrete tokens. As an illustration, BEiT (Bao et al., 2022) deploys visual tokens generated by a discrete tokenizer known as dVAE (Vahdat et al., 2018). PeCo (Dong et al., 2023) refines the token quality by introducing a tokenizer enriched with perceptual loss. More related work about MIM can be found in Appendix A. Nevertheless, we notice that different tokenization schemes may bring quite different performance. For example, as Table 1 suggests, PeCo with perceptual tokenizer and MaskFeat with HOG targets outperform MAE, whereas BEiT with dVAE and VQGAN tokenizer underperform MAE using raw pixel. These observations naturally raise the question:

*What is the role of tokenization in MIM? How does it affect the downstream generalization?*

---

*Equal Contribution. Yifei Wang has graduated from Peking University, and is currently a postdoc at MIT.
†Corresponding Author: Yisen Wang (yisen.wang@pku.edu.cn).

Table 1: Linear probing and fine-tuning accuracies (%) of several MIM methods using different tokenizers pretrained on ImageNet100 for 200 epochs. The architecture is ViT-small following Touvron et al. (2021). The comparison baseline is the tokenizer of identity function in MAE.

| Tokenizer | Linear Probing Acc. | Fine-tuning Acc. |
|---|---|---|
| Identity function (MAE) | 45.9 | 81.8 |
| dVAE (Vahdat et al., 2018) | 41.2 $(-4.7)$ | 80.2 $(-1.6)$ |
| VQGAN (Esser et al., 2021) | 44.3 $(-1.6)$ | 80.9 $(-0.9)$ |
| Perceptual tokenizer (Dong et al., 2023) | 53.2 $(+7.3)$ | 83.6 $(+1.8)$ |
| Maskfeat (HOG targets) (Wei et al., 2022) | 49.1 $(+3.2)$ | 82.8 $(+1.0)$ |
| K-means (ours) | 52.7 $(+6.8)$ | 86.4 $(+4.6)$ |

To answer these questions, we leverage the graph perspective of MAE (Zhang et al., 2022) to dissect the influence of different discrete tokenization schemes on the downstream generalization. Zhang et al. (2022) have demonstrated that masking can generate implicit positive correlations between unmasked views that share the same target masked output. However, it's highly improbable for two unmasked views to share precisely the same masked output. This discrepancy results in limited connectivity within the augmentation graph, potentially leading to suboptimal performance as shown in prior studies (Wang et al., 2022; HaoChen et al., 2021). Subsequently, we observe that when employing discrete tokens, masking creates implicit positive correlations between unmasked views as long as they share the same discrete class of target masked output. Specifically, we find that discrete tokens that align well with data classes enhance connectivity among intra-class samples, thereby improving the downstream performance. Conversely, incorrectly specified tokenizers may cause confusion between inter-class samples, resulting in poorer downstream performance.

This insight can also guide the selection of proper tokenizers. Specifically, we design a novel metric, named token-class alignment similarity (TCAS), by measuring the discrepancy in distribution between true patch labels and the discrete token labels assigned by the tokenizer. TCAS can be used to directly compare the quality of different tokenizers without training. Meanwhile, the principle inspires us to design an easy-to-implement tokenizer with its corresponding MIM method named ClusterMIM, which demonstrates its efficacy through enhanced performance across different benchmark datasets (e.g., ImageNet-100 and ImageNet-1K) and different ViT backbones.

We summarize our contributions as follows:

- We are the first to identify and theoretically formalize the role of discrete tokens in MIM, highlighting how they improve the alignment between unmasked views through the augmentation graph analysis.

- We offer a thorough theoretical exploration into the impact of discrete tokenization on the downstream generalization, emphasizing its influence on inter-class and intra-class dynamics within the augmentation graph.

- We propose a novel metric, TCAS, for evaluating tokenizer quality without training. Motivated by this metric, we propose a simple yet effective approach ClusterMIM to improve the performance of MIM methods.

## 2 THEORETICAL UNDERSTANDING OF DISCRETE TOKENIZATION IN MIM

We begin by detailing the mathematical formulation of MIM. Given a natural image, we first reshape it into $n$ patches, denoted as $\bar{x} \in \mathbb{R}^{n \times s}$ with $s$ representing the patch size. Subsequently, we employ a random binary mask $m$ drawn from the set $\{0, 1\}^n$ to create two complementary views of $\bar{x}$:

$$x_1 = \bar{x}[m] \in \mathbb{R}^{n_1 \times s}, \quad x_2 = \bar{x}[1 - m] \in \mathbb{R}^{n_2 \times s}, \tag{1}$$

where $n_1$ and $n_2$ are integers satisfying $n = n_1 + n_2$. We denote this random masking process as drawing $x_1$ and $x_2$ from the joint distribution $\mathcal{M}(x_1, x_2 | \bar{x})$ (its marginal distributions are represented as $\mathcal{M}(x_1 | \bar{x})$ and $\mathcal{M}(x_2 | \bar{x})$). Denote the set of all unmasked views as $\mathcal{X}_1 = \{x_1\}$ and the set of all masked views as $\mathcal{X}_2 = \{x_2\}$, where the two sets are assumed to be finite, i.e., $|\mathcal{X}_1| = N_1$ and

$|\mathcal{X}_2| = N_2$. The loss function for MIM can be formulated as:

$$\mathcal{L}(h) = \mathbb{E}_{\bar{x}}\mathbb{E}_{x_1,x_2|\bar{x}}L(h(x_1), t(x_2)). \qquad (2)$$

Here, the network $h$ maps the input $x_1$ to reconstruct the target $t(x_2)$. The function $t$ transforms the unmasked image patches into visual tokens, for instance, in MAE (He et al., 2022), $t$ serves as an identity function. While BEiT (Bao et al., 2022) employs a dVAE (Vahdat et al., 2018) tokenizer for $t$. The loss function $L$ can be a mean square loss form for continuous targets $t(x_2)$ (He et al., 2022; Xie et al., 2022) or a cross-entropy loss form for discrete $t(x_2)$ (Bao et al., 2022; Dong et al., 2023). In this paper, in order to simplify the analysis, we assume that $L$ takes the form of mean square loss and our primary focus revolves around distinguishing different MIM methods based on their prediction target, specifically the selection of $t$.

## 2.1 MATHEMATICAL FORMULATION OF DISCRETE TOKENIZATION

Building upon the graph-based perspective of MIM in Zhang et al. (2022), we delve into the effect of discrete tokenization on the downstream generalization. The foundational premise of Zhang et al. (2022) was to interlink $\mathcal{X}_1$ and $\mathcal{X}_2$ through a mask graph. This framework enabled them to utilize the 2-hop connectivity within the mask graph $\mathcal{G}_M$, thereby formulating an augmentation graph $\mathcal{G}_A$ to examine the relationships between element pairs in the input space $\mathcal{X}_1$. Subsequently, they derived a bound for the downstream classification error as

$$B_{downstream} \leq c_1 \sum_i \lambda_i^2 + c_2 \alpha, \qquad (3)$$

where $\lambda_i$ is the eigenvalue of normalized adjacent matrix $\bar{A}$ of augmentation graph, $\alpha$ is the mask-induced error $\alpha = \mathbb{E}_{x_1,x_1^+}\mathbb{1}[y(x_1) \neq y(x_1^+)]$, and $c_1, c_2$ are constants.

Here, our objective is to trace this analytical trajectory, specifically focusing on discerning the effects of discrete tokenization on each facet of this graph-based methodology.

**Tokenization Induces Equivalence Class in the Target Space.** In Equation 2, the function $t$ represents tokenization, which, given its intricate nature, can be analytically challenging. Therefore, it is crucial to identify the core essence of tokenization to simplify the analysis. Notably, the loss function aims to align $x_1$ and its complementary tokenized view $t(x_2)$ through the reconstruction task. Considering pairs $(x_1, x_2)$ sampled from $\mathcal{M}(\cdot, \cdot|\bar{x})$ and $(x_1^+, x_2^+)$ from $\mathcal{M}(\cdot, \cdot|\bar{x}^+)$, an intriguing observation emerges: even if $x_2 \neq x_2^+$, $x_1$ and $x_1^+$ can still share the same reconstruction target as long as $t(x_2) = t(x_2^+)$. This observation leads us to focus on the equivalence relation denoted by $\sim$ within $\mathcal{X}_2$. We define $x_2 \sim x_2^+$ if and only if $t(x_2) = t(x_2^+)$. Accordingly, we denote $\mathcal{S} = \mathcal{X}_2/\sim = \{\mathcal{S}_1, \ldots \mathcal{S}_l\}$, where each $\mathcal{S}_i$ represents an equivalence class consisting of multiple $x_2$ elements sharing the same discrete tokens. By doing so, we can treat the target view space as $\mathcal{S}$ because there will be no distinction between $x_2$ and $x_2^+$ that belong to the same $\mathcal{S}$ due to tokenization. This formalizes discrete tokenization using equivalence classes and obviates the need to delve into the specific properties of $t$. Therefore, we can equate discrete tokenization with a specific $\mathcal{S}$. Building on this, the joint probability of $x_1$ and $\mathcal{S}_i$ can be conceptualized as the summation of all joint probabilities of $x_1$ and each $x_2$ contained in $\mathcal{S}_i$:

$$\mathcal{M}(x_1, \mathcal{S}_i|\bar{x}) = \sum_{x_2 \in \mathcal{S}_i} \mathcal{M}(x_1, x_2|\bar{x}). \qquad (4)$$

This definition of the joint probability is valid, since marginalization on $\mathcal{S}_i$ makes $\mathcal{M}(x_1|\bar{x})$, i.e. $\sum_{\mathcal{S}_i \in \mathcal{S}} \mathcal{M}(x_1, \mathcal{S}_i|\bar{x}) = \sum_{\mathcal{S}_i \in \mathcal{S}} \sum_{x_2 \in \mathcal{S}_i} \mathcal{M}(x_1, x_2|\bar{x}) = \sum_{x_2 \in \mathcal{X}_2} \mathcal{M}(x_1, x_2|\bar{x}) = \mathcal{M}(x_1|\bar{x})$. It is worth noting that MAE refers to an extreme case where each $\mathcal{S}_i$ becomes a single-element set.

**Mask Graph with Tokenized Targets.** The original mask graph proposed by Zhang et al. (2022) is defined over the joint set of $\mathcal{X}_1 \cup \mathcal{X}_2$ to describe the joint probability of pairs from $\mathcal{X}_1 \times \mathcal{X}_2$. We now consider the mask graph $\mathcal{G}_M$ with tokenized targets, which is defined over the joint set $\mathcal{X} = \mathcal{X}_1 \cup \mathcal{S}$:

- Node: Each view $x_1 \in \mathcal{X}_1$ and each set of views $\mathcal{S}_i \in \mathcal{S}$.
- Edge: Edges solely exist between $x_1 \in \mathcal{X}_1$ and $\mathcal{S}_i \in \mathcal{S}$, which essentially results in a bipartite graph structure. The edge weight between $x_1$ and $\mathcal{S}_i$ is defined as their joint probability $\mathcal{M}(x_1, \mathcal{S}_i) = \mathbb{E}_{\bar{x}}\mathcal{M}(x_1, \mathcal{S}_i|\bar{x})$. Therefore, an edge with non-zero weight connects

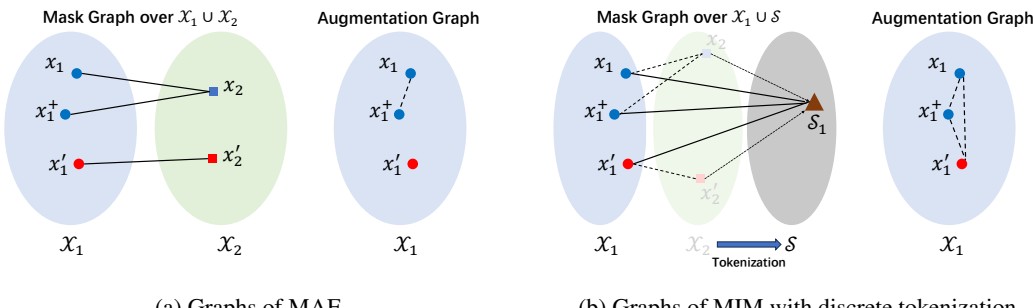

(a) Graphs of MAE $\qquad$ (b) Graphs of MIM with discrete tokenization

Figure 1: An illustration of how the discrete tokenization affects the mask graph and the corresponding augmentation graph. $x_2$ and $x_2'$ share the same discrete token, enabling a connection between $x_1$ and $x_1'$ through $x_2$ and $x_2'$, whereas such a connection is not possible in MAE.

$x_1$ and $\mathcal{S}_i$ if and only if there exists $x_2 \in \mathcal{S}_i$ such that $x_1$ and $x_2$ are complementary views generated by masking.

Figure 1 visually demonstrates the construction of the mask graph with discrete tokenization, where we can observe three pairs of complementary views: $(x_1, x_2)$, $(x_1^+, x_2)$, and $(x_1', x_2')$. Notably, since both $x_2$ and $x_2'$ belong to $\mathcal{S}_1$, this results in the formation of an edge originating from either $x_1$, $x_2$, or $x_1'$ connecting to $\mathcal{S}_1$. Intuitively, tokenization aggregates all the weights between $x_1$ and $x_2 \in \mathcal{S}_i$ into the weight between $x_1$ and $\mathcal{S}_i$. This aggregation subsequently influences the construction of the augmentation graph which will be illustrated in the following part.

**Induced Augmentation Graph.** According to Zhang et al. (2022), MAE generates implicit connections among different input samples through 2-hop connectivity in the mask graph, as it enforces a pair of 2-hop neighbor $x_1$ and $x_1^+$ to reconstruct the same output $x_2$. This treatment transforms the 2-hop input neighbors into positive pairs implicitly aligned as in contrastive learning, thereby forming the concept of an augmentation graph to model relationships among all input samples in $\mathcal{X}_1$. However, with discrete tokenization, this mechanism is modified, now requiring a 2-hop neighbor pair $x_1$ and $x_1^+$ to match the same tokenized target $\mathcal{S}_i$. In Figure 1, there is an illustration of how the discrete tokenization affects the mask graph $\mathcal{G}_M$ and the corresponding augmentation graph $\mathcal{G}_A$. Consequently, the augmentation graph influenced by discrete tokenization can be formulated as:

- Node: Each view $x_1 \in \mathcal{X}_1$ is represented as a node in $\mathcal{G}_A$.

- Edge: For any two views $x_1, x_1^+ \in \mathcal{X}_1$, we define their edge weight as the probability of having the same target view. Formally, this edge weight is computed as $\mathcal{A}(x_1, x_1^+) = \mathbb{E}_{\mathcal{S}_i} \mathcal{M}(x_1|\mathcal{S}_i)\mathcal{M}(x_1^+|\mathcal{S}_i)$.

With the augmentation graph at hand, we can analyze how the discrete tokenization influences the generalization ability, as shown in the following part.

## 2.2 DOWNSTREAM GENERALIZATION UNDER DISCRETE TOKENIZATION

In this part, we design a toy model to rigorously characterize the generalization bounds under different discrete tokenization.

**Data Space.** We have two classes, each containing $n$ points representing image patches. These point sets are denoted as $P_1$ and $P_2$. There are $m$ overlapping points between the two classes, meaning that $|P_1 \cap P_2| = m$. Therefore, there are a total of $2n - m$ points ($|P_1 \cup P_2| = 2n - m$). Assuming $t = m/n \ll 1$, we define the data distribution such that, when drawing a datum, we randomly select one class, denoted as $P_i$, and uniformly sample an ordered pair $(x_1, x_2)$ from $P_i \times P_i$.

**MIM Task and Discrete Tokenization.** In this simulated MIM task, we set $x_1$ as the unmasked view and $x_2$ as the masked view. Suppose that the equivalence class induced by the discrete tokenization is $\mathcal{S} = \{\mathcal{S}_1, \ldots, \mathcal{S}_l\}$. For the $i$-th equivalence class $\mathcal{S}_i$, it comprises $n_{i,1}$ elements from

(a) MAE-like Tokenization  (b) Class-wise Tokenization  (c) Cross-class Tokenization

Figure 2: Visual illustration of the three tokenization approaches in the toy model. Each orange bounding box represents an equivalence class, whose elements share the same discrete token. **Class-wise tokenization** exhibits a higher intra-class connectivity and lower inter-class connectivity compared to MAE-like tokenization. Consequently, it boasts a lower downstream error bound. In contrast, **cross-class tokenization** leads to lower intra-class connectivity and higher inter-class connectivity, resulting in a significantly larger downstream error bound.

$P_1/P_2$, $n_{i,2}$ elements from $P_2/P_1$ and $n_{i,3}$ elements from $P_1 \cap P_2$. Therefore, we have the following conditions: $\sum_{i=1}^{c} n_{i,1} = \sum_{i=1}^{c} n_{i,2} = n - m$, $\sum_{i=1}^{c} n_{i,3} = m$.

**Tokenizers.** We study on three kinds of tokenization: 1) MAE-like tokenization $\mathcal{S}^{\text{MAE}}$ (which essentially implies no tokenization), 2) class-wise tokenization $\mathcal{S}^{\text{class}}$, and 3) cross-class tokenization $\mathcal{S}^{\text{cross}}$. Figure 2 visually illustrates the three tokenization approaches. By calculating the weight edge and downstream error bound in each scenario, we can compare $\mathcal{S}^{\text{class}}$ and $\mathcal{S}^{\text{cross}}$ with the baseline $\mathcal{S}^{\text{MAE}}$ quantitatively. Specifically, for each tokenizer, we calculate their downstream error bounds following Equation 3 (details in Appendix B) and obtain the following results:

- **MAE-like tokenization $\mathcal{S}^{\text{MAE}}$.** In this scenario, $\mathcal{S}^{\text{MAE}} = \{\mathcal{S}_1, \ldots, \mathcal{S}_{2n-m}\}$, where each $\mathcal{S}_i$ is a single-element set similar to MAE. In this case, the edge weight between intra-class pairs $w_{\text{intra}}^{\text{MAE}}$, the edge weight between inter-class pairs $w_{\text{inter}}^{\text{MAE}}$, and the downstream error bound $B^{\text{MAE}}$ should be

$$w_{\text{intra}}^{\text{MAE}} = \frac{2n - m}{4n^3}, \ w_{\text{inter}}^{\text{MAE}} = \frac{m}{4n^3}, \ B^{\text{MAE}} = c(2 - \frac{15t}{4} + O(t^2)). \tag{5}$$

  These numerical results serve as the baselines for the other two tokenization methods.

- **Class-wise tokenization $\mathcal{S}^{\text{class}}$.** In this scenario, $\mathcal{S}^{\text{class}} = \{\mathcal{S}_1, \mathcal{S}_2\}$. The two equivalence classes divide the entire point space evenly by class, with $n_{1,1} = n_{2,2} = n - m$, $n_{1,2} = n_{2,2} = n - m$, $n_{1,3} = n_{2,3} = m/2$. In this case, the edge weight between intra-class pairs $w_{\text{intra}}^{\text{class}}$, the edge weight between inter-class pairs $w_{\text{inter}}^{\text{class}}$, and the downstream error bound $B^{\text{class}}$ should be

$$w_{\text{intra}}^{\text{class}} = \frac{(n - \frac{m}{2})^2 + (\frac{m}{2})^2}{2n^4}, \ w_{\text{inter}}^{\text{class}} = \frac{m(n - \frac{m}{2})}{4n^4}, \ B^{\text{class}} = c(2 - \frac{9t}{2} + O(t^2)). \tag{6}$$

  In comparison to MAE-like tokenization, class-wise tokenization enhances intra-class edge weights while diminishing inter-class edge weights. As suggested by HaoChen et al. (2021), this variation makes the feature space more distinct and separable by class, ultimately leading to improved downstream performance. This assertion aligns with the results of the downstream error bound calculations, where class-wise tokenization $B^{\text{class}}$ exhibits a lower downstream error bound compared to MAE-like tokenization $B^{\text{MAE}}$.

- **Cross-class tokenization $\mathcal{S}^{\text{cross}}$.** In this scenario, $n_{i,1} = n_{i,2} = (n-m)/l$ and $n_{i,3} = m/l$ which means the $l$ equivalence classes split the three sets of points equally. In this case, the edge weight between intra-class pairs $w_{\text{intra}}^{\text{cross}}$, the edge weight between inter-class pairs $w_{\text{inter}}^{\text{cross}}$, and the downstream error bound $B^{\text{cross}}$ should be

$$w_{\text{intra}}^{\text{cross}} = \frac{1}{4n^2}, \ w_{\text{inter}}^{\text{cross}} = \frac{1}{4n^2}, \ B^{\text{cross}} = c(\frac{7}{2} - \frac{27t}{4} + O(t^2)). \tag{7}$$

  In contrast to class-wise tokenization, cross-class tokenization diminishes intra-class edge weights and elevates inter-class edge weights, which, in turn, has a detrimental effect on downstream performance. This observation is further evidenced by the significantly larger downstream error bound $B^{\text{cross}}$ compared to that of MAE-like tokenization $B^{\text{MAE}}$.

**Summary.** Our findings reveal that appropriately designed discrete tokens have the potential to enhance the connectivity among intra-class samples induced by masking, leading to a better downstream performance. However, poorly chosen tokenization methods can lead to confusion among inter-class samples, resulting in a considerably larger downstream error bound that may have adverse effects on overall downstream performance.

Furthermore, we seek to uncover the characteristics of well-designed visual tokens. Given the contrast properties exhibited by $\mathcal{S}^{\text{class}}$ and $\mathcal{S}^{\text{cross}}$, we delve deeper into their specific compositions. A notable distinction emerges: each equivalence class in $\mathcal{S}^{\text{class}}$ consists of points from the same true class, while each equivalence class in $\mathcal{S}^{\text{cross}}$ comprises an equal distribution of points from both classes. This distinction leads us to wonder whether tokenization schemes aligning with the true classes result in superior downstream task performance. Affirmatively, our findings, supported by the ensuing theorem, confirm this hypothesis:

**Theorem 1.** *Assuming that $\mathcal{M}(x_1|x_2) > 0$ occurs only if $y(x_1) = y(x_2)$, and let $\sim_y$ denote the equivalence relation on $\mathcal{X}_2$ where $x_2 \sim_y x_2^+$ if and only if $y(x_2) = y(x_2^+)$. Then $\mathcal{S}^y = \mathcal{X}_2 / \sim_y = \{\mathcal{S}_1^y, \ldots, \mathcal{S}_c^y\}$ minimizes $c_1 \sum_{i=1}^{N_1} \lambda_i^2 + c_2 \alpha$.*

Theorem 1 suggests that, under the assumption that two views of an image come from the same class, the optimal discrete tokenizer is simply the label function. This insight serves as valuable inspiration for the design and selection of improved discrete tokenizers, discussed in the following.

## 3 DESIGNING AND CHOOSING BETTER DISCRETE TOKENIZER

In Section 2, we have explored the role of discrete tokens in MIM and studied their effects on the downstream generalization. Drawing from these theoretical insights, this section is dedicated to the principles of crafting and selecting more effective discrete tokenizers. We first design a metric to evaluate the discrete tokenizers in MIM. Then, based on the metric, we design a simple but effective discrete tokenizer utilizing K-means.

### 3.1 TOKEN-CLASS ALIGNMENT SIMILARITY: MEASURING THE DISCREPANCY BETWEEN TOKENIZER CLASS AND TRUE CLASS

Theorem 1 suggests that, under the assumption that two views of an image come from the same class, the optimal discrete tokenizer is simply the label function. This principle underlines our approach to devising a metric for assessing discrete tokenizers, focusing on the disparity between the equivalence classes generated by the tokenizer and those defined by the actual labels.

In practice, the set of the equivalence class $\mathcal{S}$ could be exponentially large, since there are potential $C^{n_2}$ distinct discrete representations across all masked views $\mathcal{X}_2$ where $C$ is the size of the codebook. The vast set is intractable for us to manage when designing a practical metric. Therefore, in the following part, we consider a bag-of-word model (Harris, 1954) for patch representations to make the analysis tractable. This approach considers each target view $t(x_2)$ as a collection of tokens rather than a singular entity, aligning with the localized nature of patch representations and rendering the analysis more feasible. Based on Theorem 1, we anticipate that a discrete tokenizer with a lower value of this metric will yield improved downstream performance. The metric is constructed as detailed below:

1. **Computing Token-class Co-occurrence.** To quantify the discrepancy between the tokenizer class and true class, we first establish the token-class co-occurrence matrix. Let $R \in \mathbb{R}^{l_1 \times l_2}$ be a matrix, where $l_1$ represents the size of the codebook of the discrete tokenizer, and $l_2$ represents the number of true classes. Considering all the patches in the dataset, we define $R(i, j)$ as the count of patches belonging to the $i$-th class in the codebook and the $j$-th true class. The matrix $\bar{R}$, which is the normalized version of $R$ along its rows, is given by $\bar{R}(i, j) = \text{softmax}_j(R(i, j) / \sum_{j'} R(i, j'))$. Here, the $i$-th row of the matrix $\bar{R}$, denoted as $\bar{R}(i)$, indicates the distribution of true classes within the patches that belong to the $i$-th discrete class.

2. **Measuring Tokenization-induced Class Confusion.** Ideally, if we want to minimize the discrepancy between the tokenizer class and the true class, we aim for $\bar{R}(i)$ to be close to

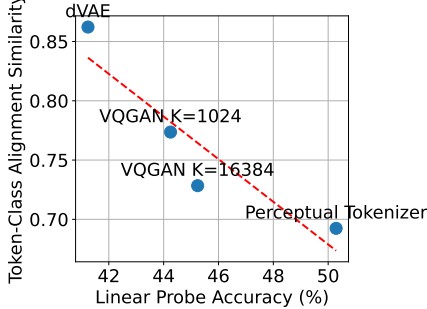

Figure 3: Correlation between TCAS and linear probing accuracy.

Table 2: Comparison of TCAS between our proposed tokenizer and previous ones.

| Method | TCAS ($\downarrow$) |
| --- | --- |
| dVAE (Vahdat et al., 2018) | 0.86 |
| VQGAN K=1024 (Esser et al., 2021) | 0.77 |
| VQGAN K=16384 (Esser et al., 2021) | 0.73 |
| Perceptual Tokenizer (Dong et al., 2023) | 0.69 |
| ClusterMIM Pixel (ours) | 0.56 |
| ClusterMIM DINO (ours) | **0.21** |

a one-hot vector, and $\bar{R}(i)$ and $\bar{R}(i')$ with $i \neq i'$ should be dissimilar. In other words, this pushes $C = \bar{R}\bar{R}^\top$ close to the identity matrix $I_{l_1 \times l_1}$. Accordingly, we formulate the metric as follows:

$$M = \lambda_1 \sum_i (1-C_{ii})^2 + \lambda_2 \sum_{i \neq i'} C_{i,i'}^2 = \lambda_1 \sum_{i=1}^{l_1} (1-\|\bar{R}(i)\|_2)^2 + \lambda_2 \sum_{i \neq i'} (\bar{R}(i)\bar{R}(i')^\top)^2. \quad (8)$$

To ensure that the metric is not influenced by the codebook size $l_1$, we set $\lambda_1 = 1/l_1$ and $\lambda_2 = 1/l_1^2$. In this metric, as $\bar{R}(i)$ approaches a one-hot vector, the first term decreases. Simultaneously, if $\bar{R}(i)$ and $\bar{R}(i')$ with $i \neq i'$ become orthogonal, the second term reaches the minimal. Thus, this metric effectively measures the discrepancy between the tokenizer class and the true class.

We designate this metric as the Token-Class Alignment Similarity (TCAS). To validate the soundness of TCAS, we conduct empirical analyses that explore its correlation with linear downstream task performance utilizing various tokenizers, including dVAE (Vahdat et al., 2018), VQGAN (Esser et al., 2021), and Perceptual tokenizer (Dong et al., 2023). The results are visually represented in Figure 3, which indicates a noticeable correlation between TCAS and performance. This significant correlation not only validates the effectiveness of TCAS as a tool for evaluating tokenizer performance but also highlights its unique advantage: enabling the assessment of tokenizers *without necessitating pretraining*. Consequently, TCAS emerges as a valuable metric, offering insightful guidance for the refinement and development of more advanced tokenization strategies.

## 3.2 CLUSTERMIM: LEVERAGING CLUSTERING FOR GENERATING DISCRETE TOKENS

Building on the insights learned from theoretical analyses and toy models, we have established that an effective tokenizer should foster label correlation, even in scenarios where explicit label information is absent —- a common scenario in self-supervised learning settings. This prompts the need for a viable strategy that can mimic label-correlated tokenization without relying on labeled data. Our above analyses suggest that the association of specific labels to discrete classes is not as crucial as ensuring that patches sharing the same labels converge within the same discrete category. Thus, we propose a novel approach that leverages clustering to produce discrete tokens.

**Tokenizer Pretraining Stage: Clustering the Patches.** Given a dataset, we begin with the collection of image patches, akin to the inputs used by ViT (Dosovitskiy et al., 2020). Then we employ a clustering algorithm to group these patches, yielding a set of $K$ clustering centers. These centers constitute the codebook for the discrete tokenizer.

**Tokenizer Inference Stage: Assigning the Nearest Neighbor.** To determine the discrete tokens for an image patch, we identify the nearest neighbor in the codebook, which is then designated as the discrete token for the respective patch.

We term the MIM approach using this tokenizer as ClusterMIM (Clustering Masked Image Modeling), which bifurcates into two practical implementations based on the clustering method: 1) ClusterMIM Pixel, which directly uses K-means in the pixel space, and 2) ClusterMIM DINO, which

uses K-means in the feature space generated by DINO (Caron et al., 2021). Preliminary results, as showcased in Table 2, reveal that our methods achieve a notable reduction in token-class alignment similarity (TCAS) compared to baselines, underscoring the potential of our clustering-based tokenizer. The following experiments, detailed in Section 4, further validate the efficacy of Cluster-MIM, providing strong empirical evidence for our proposed approach.

## 4 EXPERIMENTS

In this section, we first present the main empirical results of our proposed ClusterMIM methods on different real-world datasets with different backbones. Then we conduct a series of ablation experiments to discuss the selection of hyperparameters in ClusterMIM.

### 4.1 EVALUATION ON BENCHMARK DATASETS

To evaluate the effectiveness of the proposed ClusterMIM method, extensive experiments are conducted on ImageNet-100 (Deng et al., 2009) and ImageNet-1K (Deng et al., 2009).

**Setup.** We mainly follow the basic setup of MAE (He et al., 2022): for the encoder, we adopt several variants of ViT (Dosovitskiy et al., 2020), including ViT-Small, ViT-Base, and ViT-Large. For the decoder, we follow the setting of He et al. (2022). The mask ratio is set to 0.75. On both datasets, we pretrain the model for 200 epochs with batch size 4096 and weight decay 0.05. We conduct both linear evaluation and non-linear fine-tuning on the pretrained encoder. For linear evaluation, we train a linear classifier on the frozen pretrained encoder. As for non-linear fine-tuning, we train both the pretrained encoder and the linear classifier with the soft target cross entropy loss (Peterson et al., 2019). For the K-Means algorithm used in the tokenizer pretraining stage, we use K-Means++ initialization (Arthur & Vassilvitskii, 2007). We train K-Means for 100 epochs on ImageNet-100 and 10 epochs on ImageNet-1K.

**Effectiveness of the Proposed ClusterMIM Method.** In Table 3, we present a performance comparison of different MIM methods across two benchmark datasets. Notably, our proposed methods exhibit a significant performance advantage over all baseline methods. Specifically, on ImageNet-100, with minimal extra time constructing discrete tokens through K-Means on the pixel space (5 hours on K-Means and 36 hours on pretraining), our ClusterMIM Pixel utilizing the ViT-S backbone outperform MAE by an impressive 6.8% in linear probing and 4.6% in fine-tuning accuracy. The improvement still holds when using larger backbones and larger ImageNet-1K dataset.

Table 3: Linear probing accuracy (LP Acc.) and fine-tuning accuracy (FT Acc.) of pretrained models by various MIM methods with different ViT backbones on ImageNet-100 and ImageNet-1K. ViT-S, ViT-B and ViT-L are abbreviations of ViT-Small, ViT-Base and ViT-Large, repectively. **Bold** indicates the best performance within the same setting.

| Dataset | Backbone | Method | Extra Model | LP Acc. | FT Acc. |
|---|---|---|---|---|---|
| ImageNet-100 | ViT-S | MAE (He et al., 2022) | No | 45.9 | 81.8 |
| | | MaskFeat HOG ver.(Wei et al., 2022) | No | 49.1 | 82.8 |
| | | ClusterMIM Pixel (ours) | No | **52.7** | **86.4** |
| | | BEiT (Bao et al., 2022) | dVAE | 43.2 | 81.0 |
| | | PeCo (Dong et al., 2023) | VQVAE | 53.2 | 83.6 |
| | | MaskFeat continuous ver. (Wei et al., 2022) | DINO | 57.1 | 84.3 |
| | | ClusterMIM DINO (ours) | DINO | **59.7** | **84.7** |
| | ViT-B | MAE (He et al., 2022) | No | 61.2 | 86.9 |
| | | BEiT (Bao et al., 2022) | dVAE | 55.8 | 86.1 |
| | | ClusterMIM Pixel (ours) | No | **63.1** | **88.8** |
| | ViT-L | MAE (He et al., 2022) | No | 64.4 | 87.3 |
| | | ClusterMIM Pixel (ours) | No | **67.2** | **88.9** |
| ImageNet-1K | ViT-B | MAE (He et al., 2022) | No | 55.4 | 82.9 |
| | | ClusterMIM Pixel (ours) | No | **62.1** | **83.2** |
| | | BEiT (Bao et al., 2022) | dVAE | 53.2 | 82.7 |
| | | MaskFeat continuous ver. (Wei et al., 2022) | DINO | 63.2 | 83.7 |
| | | ClusterMIM DINO (ours) | DINO | **67.4** | **83.8** |

Table 4: Ablation study on the selection of clustering number $K$. Linear probing accuracy / Fine-tuning accuracy in each box.

| $K$ | ClusterMIM Pixel | ClusterMIM DINO |
|------|------------------|-----------------|
| 50 | **52.7/86.4** | 58.2/84.3 |
| 100 | 50.1/83.7 | **59.7**/84.7 |
| 1000 | 50.6/84.4 | 56.9/**85.1** |

Table 5: Experiments exploring different K-Means training epochs. Linear probing accuracy / Fine-tuning accuracy in each box.

| Epoch | ClusterMIM Pixel | ClusterMIM DINO |
|-------|------------------|-----------------|
| 1 | 47.2/82.1 | 53.5/84.3 |
| 10 | 52.5/85.9 | 57.4/84.4 |
| 100 | **52.7/86.4** | **59.7/84.7** |

Furthermore, when using the discrete tokens generated by pretrained DINO features as the reconstruction target (ClusterMIM DINO) instead of the pretrained DINO features themselves (MaskFeat continuous version), we achieve a performance boost of 2.6% in linear probing and 0.4% in fine-tuning accuracy. These improvements provide direct evidence of the effectiveness of our proposed tokenizer.

## 4.2 ABLATION STUDY

In this section, we conduct an ablation study to investigate the impact of two key factors in our proposed ClusterMIM method: the choice of clustering number $K$ and the training epochs of the K-Means algorithm. These experiments are conducted on the ImageNet-100 dataset using the ViT-small architecture.

**Clustering Number $K$.** We vary the clustering number $K$ for the K-Means algorithm while fixing the training epoch to be 100. The results of these ablation experiments are presented in Table 4. As observed in the table, different choices of clustering numbers have varying effects on the downstream performance. In the case of ClusterMIM Pixel, we achieve the highest linear probing accuracy of 52.7% and fine-tuning accuracy of 86.4% when $K = 50$. Conversely, for ClusterMIM DINO, the best performance is achieved when $K = 100$, yielding linear probing accuracy of 59.7% and fine-tuning accuracy of 84.7%. Notably, using a very large $K = 1000$ did not necessarily lead to improved performance. Therefore, it is advisable to select a moderate clustering number in practice.

**Training Epochs in K-Means.** We conducted experiments to explore the influence of varying the training epochs for the K-Means algorithm within the ClusterMIM framework. The results of these experiments are presented in Table 5. Across all scenarios, the highest accuracies are consistently achieved when training K-Means for 100 epochs, demonstrating the importance of a well-trained K-Means model in enhancing ClusterMIM performance. It is noteworthy that the K-Means algorithm is computationally efficient. For instance, in the case of ClusterMIM Pixel trained on ImageNet-100 with 200 epochs, while the pretraining stage takes 36 hours, training 1 epoch of K-Means with 100 cluster centers requires only 0.05 hours. Furthermore, it's worth highlighting that even with just 1 epoch of K-Means pretraining, ClusterMIM Pixel outperformed MAE (47.2/82.1 vs. 45.9/81.8), and ClusterMIM DINO outperformed the continuous version of MaskFeat (53.5/84.3 vs. 53.1/84.3). These results underscore the effectiveness and efficiency of our proposed methods.

## 5 CONCLUSION

In this paper, we draw attention to the crucial role of discrete tokens in the efficacy of MIM techniques. We introduce the first in-depth theoretical analysis dedicated to understanding the impact of various discrete tokenization strategies on MIM methods. Through a nuanced examination from the mask graph perspective, we have unveiled how discrete tokenization critically influences downstream model performance. Central to our findings is the revelation that tokenizer alignment with actual labels is instrumental in enhancing model efficacy. Building upon these insights, we develop Token-Class Alignment Similarity (TCAS), a novel metric to quantitatively assess the quality of a given tokenizer. Leveraging this metric, we further propose a straightforward yet effective discrete tokenizer along with an accompanying MIM approach, termed ClusterMIM. Our empirical evaluations across a range of benchmark datasets demonstrate the superior performance of our proposed ClusterMIM, highlighting its effectiveness in enhancing model efficacy through improved discrete tokenization. Hope our work could inspire some new tokenizer designs in MIM methods.

## ACKNOWLEDGEMENT

Yisen Wang was supported by National Key R&D Program of China (2022ZD0160300), National Natural Science Foundation of China (62376010, 92370129), Beijing Nova Program (20230484344), and CCF-BaiChuan-Ebtech Foundation Model Fund.

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

## A    ADDITIONAL RELATED WORKS

**Masked Image Modeling (MIM).** MIM involves learning from a portion of masked input signals by predicting these signals using the unmasked part. Different masked prediction objectives have been proposed for MIM. For instance, MAE (He et al., 2022), and its variants like U-MAE (Zhang et al., 2022) and MixMAE (Liu et al., 2023), directly predict the raw pixels within the masked part. DBot (Zhou et al., 2023) can also use representation from another pretrained model as prediction target, but updating the target per certain epochs with current model representation. Notably, the current designs of discrete tokens in these methods do not possess theoretical guarantees. In this paper, we provide a theoretical understanding of the working mechanism of discrete tokens in MIM, especially regarding their impact on downstream generalization.

**Theoretical Understanding of MIM.** Despite the remarkable success of MIM methods, their theoretical understandings remain rarely explored. Cao et al. (2022) focus on the attention mechanism of MAE through an integral kernel perspective. Zhang et al. (2022) build a connection between MAE and contrastive learning, highlighting the importance of masking. While these theoretical approaches offer meaningful insights into MAE, they sidestep an examination of the influence of discrete tokens on MIM. Therefore, our work intends to explore the specific role and impact of discrete tokens within the MIM framework.

## B    DETAILED CALCULATION IN SECTION 2.2

We denote $w_{x_1,x_2} = \mathcal{M}(x_1, x_2)$ as the joint probability of one complementary pairs and denote $w_{x_1} = \mathcal{M}(x_1)$ as the marginal probability of one view.

Consider the joint probability of one complementary pair $x_1, x_2$: When the two views both come from the same overlapped region, the joint probability $w_{x_1,x_2} = 1/n^2$. Otherwise, the joint probability is $1/2n^2$.

For the masked view $x_2$ or unmasked view $x_1$ in the non-overlapped region, the marginal probability is $1/(2n)$. For masked views and unmasked views in the overlapped region, the marginal probability is $1/n$.

The normalized augmentation graph is

$$\bar{A}_{\text{discrete}}(x_1, x_1^+) = \sum_{x_2, x_2^+} \frac{w_{x_1,x_2} w_{x_1^+,x_2^+} \cdot w_{x_2,x_2^+}}{w_{x_2} w_{x_2^+} \sqrt{w_{x_1} w_{x_1^+}}} = \sum_{\mathcal{D}_i} \sum_{x_2, x_2^+ \in \mathcal{D}_i} \frac{w_{x_1,x_2} w_{x_1^+,x_2^+}}{\sqrt{w_{x_1} w_{x_1^+}} p(\mathcal{D}_i)}. \quad (9)$$

We calculate the joint probability of a positive pair under each situation. Each item denotes where the two unmasked views of the pair come from:

- $(P_1/P_2, P_1/P_2)$

$$\begin{aligned}
p_{11} &= 2n \cdot \sum_{i=1}^{l} \frac{2n}{n_{i,1} + n_{i,2} + 2n_{i,3}} \cdot \left( (n_{i,1} + n_{i,3})^2 \cdot \frac{1}{2n^2} \cdot \frac{1}{2n^2} \right) \\
&= \sum_{i=1}^{l} \frac{1}{n_{i,1} + n_{i,2} + 2n_{i,3}} \cdot \left( (n_{i,1} + n_{i,3})^2 \cdot \frac{1}{n^2} \right)
\end{aligned} \quad (10)$$

- $(P_2/P_1, P_2/P_1)$

$$\begin{aligned}
p_{22} &= 2n \cdot \sum_{i=1}^{l} \frac{2n}{n_{i,1} + n_{i,2} + 2n_{i,3}} \cdot \left( (n_{i,2} + n_{i,3})^2 \cdot \frac{1}{2n^2} \cdot \frac{1}{2n^2} \right) \\
&= \sum_{i=1}^{l} \frac{1}{n_{i,1} + n_{i,2} + 2n_{i,3}} \cdot \left( (n_{i,2} + n_{i,3})^2 \cdot \frac{1}{n^2} \right)
\end{aligned} \quad (11)$$

- $(P_1/P_2, P_2/P_1)$

$$p_{12} = 2n \cdot \sum_{i=1}^{l} \frac{2n}{n_{i,1} + n_{i,2} + 2n_{i,3}} \cdot \left( (n_{i,1} + n_{i,3})(n_{i,2} + n_{i,3}) \cdot \frac{1}{2n^2} \cdot \frac{1}{2n^2} \right)$$

$$= \sum_{i=1}^{l} \frac{1}{n_{i,1} + n_{i,2} + 2n_{i,3}} \cdot \left( n_{i,3}^2 \cdot \frac{1}{n^2} \right) \quad (12)$$

- $(P_1/P_2, P_1 \cap P_2)$

$$p_{13} = \sqrt{2}n \cdot \sum_{i=1}^{l} \frac{2n}{n_{i,1} + n_{i,2} + 2n_{i,3}} \cdot \left( (n_{i,1} + n_{i,3})n_{i,1} \cdot \frac{1}{2n^2} \cdot \frac{1}{2n^2} + (n_{i,1} + n_{i,3})n_{i,3} \cdot \frac{1}{n^2} \cdot \frac{1}{2n^2} \right)$$

$$= \sum_{i=1}^{l} \frac{1}{n_{i,1} + n_{i,2} + 2n_{i,3}} \cdot \left( (n_{i,1} + n_{i,3})(n_{i,1} + 2n_{i,3}) \cdot \frac{\sqrt{2}}{2n^2} \right) \quad (13)$$

- $(P_2/P_1, P_1 \cap P_2)$

$$p_{23} = \sqrt{2}n \cdot \sum_{i=1}^{l} \frac{2n}{n_{i,1} + n_{i,2} + 2n_{i,3}} \cdot \left( (n_{i,2} + n_{i,3})n_{i,2} \cdot \frac{1}{2n^2} \cdot \frac{1}{2n^2} + (n_{i,2} + n_{i,3})n_{i,3} \cdot \frac{1}{n^2} \cdot \frac{1}{2n^2} \right)$$

$$= \sum_{i=1}^{l} \frac{1}{n_{i,1} + n_{i,2} + 2n_{i,3}} \cdot \left( (n_{i,2} + n_{i,3})(n_{i,2} + 2n_{i,3}) \cdot \frac{\sqrt{2}}{2n^2} \right) \quad (14)$$

- $(P_1 \cap P_2, P_1 \cap P_2)$

$$p_{33} = n \cdot \sum_{i=1}^{l} \frac{2n}{n_{i,1} + n_{i,2} + 2n_{i,3}} \cdot \left( (n_{i,1} + n_{1,2} + 2n_{i,3})^2 \cdot \frac{1}{2n^2} \cdot \frac{1}{2n^2} \right)$$

$$= \sum_{i=1}^{l} \frac{1}{n_{i,1} + n_{i,2} + 2n_{i,3}} \cdot \left( (n_{i,1} + n_{1,2} + 2n_{i,3})^2 \cdot \frac{1}{2n^2} \right) \quad (15)$$

$$= \frac{1}{n}$$

The normalized adjacent matrix is:

$$A = \begin{bmatrix}
p_{11} & p_{11} & p_{11} & \cdots & p_{12} & p_{12} & p_{12} & \cdots & p_{13} \\
p_{11} & p_{11} & p_{11} & \cdots & p_{12} & p_{12} & p_{12} & \cdots & p_{13} \\
p_{11} & p_{11} & p_{11} & \cdots & p_{12} & p_{12} & p_{12} & \cdots & p_{13} \\
\cdots & \cdots & \cdots & \cdots & \cdots & \cdots & \cdots & \cdots & \cdots \\
p_{12} & p_{12} & p_{12} & \cdots & p_{22} & p_{22} & p_{22} & \cdots & p_{23} \\
p_{12} & p_{12} & p_{12} & \cdots & p_{22} & p_{22} & p_{22} & \cdots & p_{23} \\
p_{12} & p_{12} & p_{12} & \cdots & p_{22} & p_{22} & p_{22} & \cdots & p_{23} \\
\cdots & \cdots & \cdots & \cdots & \cdots & \cdots & \cdots & \cdots & \cdots \\
p_{13} & p_{13} & p_{13} & \cdots & p_{23} & p_{23} & p_{23} & \cdots & p_{33}
\end{bmatrix} \quad (16)$$

The sum of square of the eigenvalues is equal to the sum of square of the elements in the adjacent matrix:

$$\sum_i \lambda_i^2 = (n-m)^2(p_{11}^2 + p_{22}^2 + 2p_{12}^2) + (n-m)m(p_{13}^2 + p_{23}^2) + m^2 p_{33}^2$$

$$= \frac{(n-m)^2}{n^4} \left( (\sum_{i=1}^{l} \frac{(n_{i,1} + n_{i,3})^2}{n_{i,1} + n_{i,2} + 2n_{i,3}})^2 + (\sum_{i=1}^{l} \frac{(n_{i,2} + n_{i,3})^2}{n_{i,1} + n_{i,2} + 2n_{i,3}})^2 + 2(\sum_{i=1}^{l} \frac{(n_{i,1} + n_{i,3})(n_{i,2} + n_{i,3})}{n_{i,1} + n_{i,2} + 2n_{i,3}})^2 \right)$$

$$+ \frac{(n-m)m}{2n^4} \left( (\sum_{i=1}^{l} \frac{(n_{i,1} + n_{i,3})(n_{i,1} + 2n_{i,3})}{n_{i,1} + n_{i,2} + 2n_{i,3}})^2 + (\sum_{i=1}^{l} \frac{(n_{i,2} + n_{i,3})(n_{i,2} + 2n_{i,3})}{n_{i,1} + n_{i,2} + 2n_{i,3}})^2 \right) + \frac{m^2}{n^2} \quad (17)$$

The labeling error should be

$$\alpha = \mathbb{E}_{x_1, x_1^+} 1_{y(x_1) \neq y(x_1^+)} = 2(n-m)^2 p_{12}/(2n) = \frac{(n-m)^2}{n^3} \sum_{i=1}^{l} \frac{(n_{i,1} + n_{i,3})(n_{i,2} + n_{i,3})}{n_{i,1} + n_{i,2} + 2n_{i,3}} \quad (18)$$

We consider the downstream error bound derived in (Zhang et al., 2022), which is

$$
\begin{aligned}
B &= c_1 \sum_i \lambda_i^2 + c_2 \alpha \\
&= c \left( \frac{(n-m)^2}{n^4} \left( (\sum_{i=1}^{l} \frac{(n_{i,1} + n_{i,3})^2}{n_{i,1} + n_{i,2} + 2n_{i,3}})^2 + (\sum_{i=1}^{l} \frac{(n_{i,2} + n_{i,3})^2}{n_{i,1} + n_{i,2} + 2n_{i,3}})^2 + 2(\sum_{i=1}^{l} \frac{(n_{i,1} + n_{i,3})(n_{i,2} + n_{i,3})}{n_{i,1} + n_{i,2} + 2n_{i,3}})^2 \right) \right) \\
&\quad + \frac{(n-m)m}{2n^4} \left( (\sum_{i=1}^{l} \frac{(n_{i,1} + n_{i,3})(n_{i,1} + 2n_{i,3})}{n_{i,1} + n_{i,2} + 2n_{i,3}})^2 + (\sum_{i=1}^{l} \frac{(n_{i,2} + n_{i,3})(n_{i,2} + 2n_{i,3})}{n_{i,1} + n_{i,2} + 2n_{i,3}})^2 \right) + \frac{m^2}{n^2} + c_3 \alpha \right) \\
&= c \left( \frac{(n-m)^2}{n^4} \left( (\sum_{i=1}^{l} \frac{(n_{i,1} + n_{i,3})^2}{n_{i,1} + n_{i,2} + 2n_{i,3}})^2 + (\sum_{i=1}^{l} \frac{(n_{i,2} + n_{i,3})^2}{n_{i,1} + n_{i,2} + 2n_{i,3}})^2 + 2(\sum_{i=1}^{l} \frac{(n_{i,1} + n_{i,3})(n_{i,2} + n_{i,3})}{n_{i,1} + n_{i,2} + 2n_{i,3}} + \frac{c_3 n}{4})^2 \right) \right) \\
&\quad + \frac{(n-m)m}{2n^4} \left( (\sum_{i=1}^{l} \frac{(n_{i,1} + n_{i,3})(n_{i,1} + 2n_{i,3})}{n_{i,1} + n_{i,2} + 2n_{i,3}})^2 + (\sum_{i=1}^{l} \frac{(n_{i,2} + n_{i,3})(n_{i,2} + 2n_{i,3})}{n_{i,1} + n_{i,2} + 2n_{i,3}})^2 \right) + \frac{m^2}{n^2} - C \right).
\end{aligned}
\quad (19)
$$

where $C = c_3^2 (n-m)^2/(8n^2)$ and $c_3 = 5/2$. We denote $t = m/n$ and assume $t \ll 1$. Since $p_{11}, p_{12}$ and $p_{22}$ are elements from normalized adjacency matrix, the edge weight of intra-class pairs can be computed as $(p_{11} + p_{22})/4n$ and the edge weight of inter-class pairs can be computed as $p_{12}/4n$.

**MAE Tokenization.** We have $l = 2n - m$ and only one of $n_{i,1}$, $n_{i,2}$ and $n_{i,3}$ is 1 and the other two are 0 for each $1 \leq i \leq l$. The bound for MAE tokenization should be

$$
\begin{aligned}
B^{\text{MAE}} &= c \left( \frac{(n-m)^2}{n^4} (2(n - \frac{m}{2})^2 + 2(\frac{m}{2} + \frac{cn}{4})^2) + \frac{(n-m)m}{2n^4}(n^2 + n^2) + \frac{m^2}{n^2} - C \right) \\
&= c(2 - 5t + 7t^2 - 4t^3 + t^4 + \frac{c}{2}(1-t)^2 t) \\
&= c(2 - \frac{15}{4}t + \frac{9}{2}t^2 - \frac{11}{4}t^3 + t^4)
\end{aligned}
\quad (20)
$$

The two types of edge weights are

$$w_{\text{intra}}^{\text{MAE}} = \frac{2n - m}{4n^3}, \; w_{\text{inter}}^{\text{MAE}} = \frac{m}{4n^3} \quad (21)$$

**Class-wise Tokenization.** We consider an extreme case where $l = 2, n_{1,1} = n - m, n_{1,2} = 0, n_{1,3} = m/2$ and $n_{2,1} = 0, n_{2,2} = n - m, n_{2,3} = m/2$. The bound for this case should be

$$
\begin{aligned}
B^{\text{class}} &= c(2 - 7t + 16t^2 - \frac{39}{2}t^3 + \frac{29}{2}t^4 - \frac{95}{16}t^5 + \frac{15}{16}t^6 + c\frac{(n-m)^2 m(n - \frac{m}{2})}{n^4}) \\
&= c(2 - 7t + 16t^2 - \frac{39}{2}t^3 + \frac{29}{2}t^4 - \frac{95}{16}t^5 + \frac{15}{16}t^6 + \frac{5}{2}(1-t)^2(t - \frac{t^2}{2})) \\
&= c(2 - \frac{9}{2}t + \frac{39}{4}t^2 + O(t^3)).
\end{aligned}
\quad (22)
$$

The two types of edge weights are

$$w_{\text{intra}}^{\text{class}} = \frac{(n - \frac{m}{2})^2 + (\frac{m}{2})^2}{2n^4}, \; w_{\text{inter}}^{\text{class}} = \frac{m(n - \frac{m}{2})}{4n^4} \quad (23)$$

**Cross-class Tokenization.** Here, $n_{i,1} = n_{i,2} = (n-m)/l$ and $n_{i,3} = m/l$. The bound should be

$$
\begin{aligned}
B^{\text{class}} &= c(\frac{(n-m)^2}{n^4}(\frac{n^2}{4} + \frac{n^2}{4} + 2\frac{n^2}{4}) + \frac{(n-m)m}{2n^4}((\frac{n+m}{2})^2 + (\frac{n+m}{2})^2) + \frac{m^2}{)}n^2 + c\frac{(n-m)^2}{2n^2} \\
&= c((1-t)^2 + \frac{1}{4}(1-t)t(t+1)^2 + t^2 + \frac{c}{2}(1-t)^2) \\
&= c(\frac{7}{2} - \frac{27}{4}t + \frac{15}{4}t^2 + O(t^3)).
\end{aligned}
$$
(24)

The two types of edge weights are

$$
w_{\text{intra}}^{\text{cross}} = \frac{1}{4n^2}, \ w_{\text{inter}}^{\text{cross}} = \frac{1}{4n^2},
$$
(25)

## C Toy Model with Multiple Classes

In this section, We extend the discussion in Section 2.2 from two classes to multiple classes. Here are the setting of the toy models.

- **Data Space.** We have $s$ classes, each containing $n$ points representing image patches. These point sets are denoted as $P_i$, $i = 1, 2, \ldots s$. There are $m$ overlapping points between any two classes, meaning that $|P_i \cap P_j| = m$. There does not exist any point belonging to three classes. Therefore, there are a total of $sn - ms(s-1)/2$ points. Assuming $t = m/n \ll 1$, we define the data distribution such that, when drawing a datum, we randomly select one class, denoted as $P_i$, and uniformly sample an ordered pair $(x_1, x_2)$ from $P_i \times P_i$.

- **MIM Task and Discrete Tokenization.** In this simulated MIM task, we set $x_1$ as the unmasked view and $x_2$ as the masked view. Suppose that the equivalence class induced by the discrete tokenization is $\mathcal{S} = \{\mathcal{S}_1, \ldots, \mathcal{S}_l\}$. For the $i$-th equivalence class $\mathcal{S}_i$, it comprises $n_{i,a,b}$ elements from $P_a \cap P_b$ where $a \neq b$, $n_{i,a}$ elements from $P_a / \cup_{b \neq a} P_b$. Therefore, we have the following conditions:

$$
\sum_{i=1}^{c} n_{i,a} = n - (s-1) * m, \ \sum_{i=1}^{c} n_{i,a,b} = m, \ a \neq b.
$$
(26)

**Tokenizers.** We also study on three kinds of tokenization: MAE-like tokenization $\mathcal{S}^{\text{MAE}}$ (which essentially implies no tokenization), class-wise tokenization $\mathcal{S}^{\text{class}}$ and cross-class tokenization $\mathcal{S}^{\text{cross}}$. By calculating the weight edge and downstream error bound in each scenario, we compare $\mathcal{S}^{\text{class}}$ and $\mathcal{S}^{\text{cross}}$ with the baseline $\mathcal{S}^{\text{MAE}}$. By doing this, we are able to explore how tokenization influences the augmentation graph and the downstream error bound as follows:

- **MAE-like $\mathcal{S}^{\text{MAE}}$.** In this scenario, $\mathcal{S}^{\text{MAE}} = \{\mathcal{S}_1, \ldots, \mathcal{S}_{sn-ms(s-1)/2}\}$, where each $\mathcal{S}_i$ is a single-element set similar to MAE. In this case, the edge weight between intra-class pairs $w_{\text{intra}}^{\text{MAE}}$, the edge weight between inter-class pairs $w_{\text{inter}}^{\text{MAE}}$, the downstream error bound $B^{\text{MAE}}$ should be

$$
\begin{aligned}
w_{\text{intra}}^{\text{MAE}} &= \frac{2n - (s-1)m}{2sn^3}, \ w_{\text{inter}}^{\text{MAE}} = \frac{m}{2sn^3}, \\
B^{\text{MAE}} &= c(s - \frac{15(s-1)t}{4} + O(t^2)).
\end{aligned}
$$
(27)

  These numerical results serve as the baselines for the other two tokenization methods.

- **Class-wise $\mathcal{S}^{\text{class}}$.** In this scenario, $\mathcal{S}^{\text{class}} = \{\mathcal{S}_1, \mathcal{S}_2, \ldots, \mathcal{S}_s\}$. The $s$ equivalence classes divide the entire point space evenly by class, with $n_{s,s,b} = m/2$, $n_{s,s} = n - (s-1)m$. In this case, the edge weight between intra-class pairs $w_{\text{intra}}^{\text{class}}$, the edge weight between inter-class pairs $w_{\text{inter}}^{\text{class}}$, the downstream error bound $B^{\text{class}}$ should be

$$
\begin{aligned}
w_{\text{intra}}^{\text{class}} &= \frac{(n - (s-1)\frac{m}{2})^2 + (s-1)(\frac{m}{2})^2}{2sn^4}, \ w_{\text{inter}}^{\text{class}} = \frac{m(n - (s-1)\frac{m}{2})}{2sn^4}, \\
B^{\text{class}} &= c(s - \frac{9(s-1)t}{2} + O(t^2)).
\end{aligned}
$$
(28)

In comparison to MAE-like tokenization, class-wise tokenization enhances intra-class edge weights while diminishing inter-class edge weights. This leads to lower downstream error bound, which is consistent with the situation where $s = 2$.

– **Cross-class $\mathcal{S}^{\text{cross}}$.** In this scenario, $n_{i,a} = (n-(s-1)m)/l$ and $n_{i,a,b} = m/l$, , $a \neq b$ which means the $l$ equivalence classes split each set of points equally. In this case, the edge weight between intra-class pairs $w_{\text{intra}}^{\text{cross}}$, the edge weight between inter-class pairs $w_{\text{inter}}^{\text{cross}}$, the downstream error bound $B^{\text{cross}}$ should be

$$w_{\text{intra}}^{\text{cross}} = \frac{1}{2sn^2}, \ w_{\text{inter}}^{\text{cross}} = \frac{1}{2sn^2}, \ B^{\text{cross}} = c(\frac{1}{2} + \frac{3s}{2} + O(t)). \tag{29}$$

In contrast to class-wise tokenization, cross-class tokenization diminishes intra-class edge weights and elevates inter-class edge weights, which, in turn, has a detrimental effect on downstream performance. This observation is further evidenced by the significantly larger downstream error bound $B^{\text{cross}}$ compared to that of MAE-like tokenization $B^{\text{MAE}}$.

In conclusion, the conclusion for multiple classes is similar to that for two classes. While class-wise tokenization will obtain larger intra-class connectivity and lower inter-class connectivity, leading to lower downstream error bound and potentially better downstream performance. On the other hand, cross-class tokenization that does not align well with the class, obtain larger inter-class connectivity and lower intra-class connectivity, leading to larger downstream error bound.

## D  EXPERIMENTS ON LONGER TRAINING

We conduct additional experiments using ClusterMIM Pixel involving 800 epochs of pretraining on ImageNet-1k using ViT-B as the backbone. The comparisons with baseline methods are presented in Table 6. Notably, with only half the number of pretraining epochs, ClusterMIM Pixel reaches the same finetuning accuracy with MAE and ourperforms MAE in the linear probing accuracy. The results indicate that ClusterMIM Pixel outperforms the baseline results in the classification tasks, ensuring the better learning ability of ClusterMIM.

Table 6: Linear probing accuracy and fine-tuning accuracy of pretrained models with ViT-B on ImageNet-1K. Bold indicates the best performance.

| Methods | Epochs | Linear Probing Accuracy | Fine-tuning Accuracy |
|---|---|---|---|
| MAE | 1600 | 68.0 | **83.6** |
| BEiT | 800 | N/A | 83.2 |
| ClusterMIM Pixel | 800 | **69.4** | **83.6** |

## E  TCAS SCORE OF VARIOUS TOKENIZERS

In this section, we will present and analyze the TCAS scores of various tokenizers and the corresponding performance on ImageNet-100. The results are detailed in Table 7, and we visually organize the outcomes in Figure 4, where we use linear regression to fit the data. The r-value of the fitting is 0.85, demonstrating a strong correlation between TCAS score and linear probing accuracy. Specifically, the tokenizer's linear probing performance tends to improve as the TCAS metric decreases. Therefore, TCAS can serve as a valuable guidance for assessing the quality of a tokenizer.

Table 7: TCAS scores and performances on classification tasks of various discrete tokenizers on ImageNet-100 with 200 epochs pretraining.

| $K$ | Method | TCAS | Linear Probing Acc. | Fine-tuning Acc. |
|---|---|---|---|---|
| 10 | ClusterMIM Pixel | 0.69 | 42.7 | 81.9 |
| 25 | ClusterMIM Pixel | 0.60 | 48.3 | 84.6 |
| 50 | ClusterMIM Pixel (KMEANS 1 epoch) | 0.66 | 47.2 | 82.1 |
| 50 | ClusterMIM Pixel (KMEANS 10 epochs) | 0.56 | 52.5 | 85.9 |
| 50 | ClusterMIM Pixel (KMEANS 100 epochs) | 0.56 | 52.7 | 86.4 |
| 50 | ClusterMIM DINO | 0.21 | 58.2 | 84.3 |
| 100 | ClusterMIM Pixel | 0.52 | 50.1 | 83.7 |
| 100 | ClusterMIM MAE | 0.34 | 52.3 | 83.2 |
| 100 | ClusterMIM DINO | 0.15 | 59.7 | 84.7 |
| 100 | ClusterMIM DeiT | 0.09 | 64.8 | 83.1 |
| 200 | ClusterMIM Pixel | 0.49 | 50.8 | 84.8 |
| 1000 | ClusterMIM Pixel | 0.39 | 50.6 | 84.4 |
| 1000 | ClusterMIM DINO | 0.12 | 56.9 | 85.1 |
| 1024 | VQGAN | 0.77 | 44.3 | 80.9 |
| 8192 | dVAE | 0.86 | 41.2 | 80.2 |
| 8192 | Perceptual Tokenizer | 0.69 | 50.3 | 83.1 |

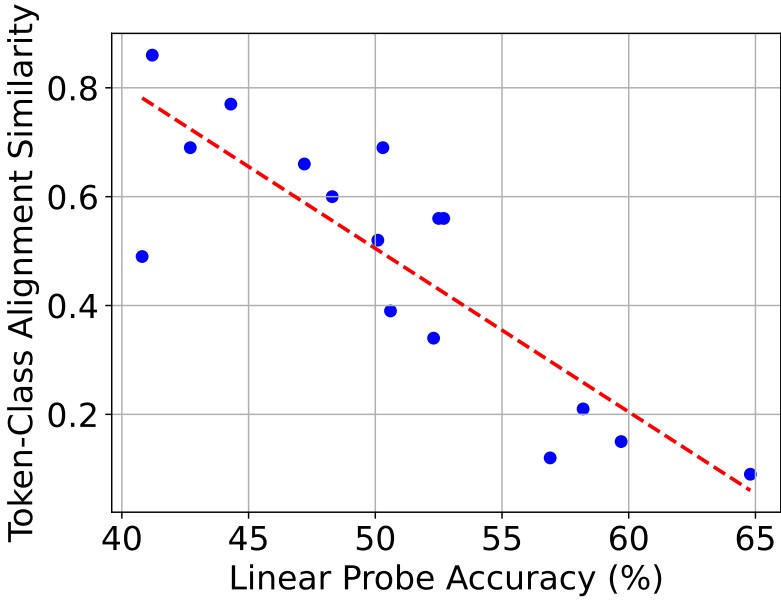

Figure 4: The relation between TCAS score and linear probing accuracy on ImageNet-100.

