# OpenReview forum: "On the Role of Discrete Tokenization in Visual Representation Learning"
_ICLR.cc/2024/Conference — ICLR 2024 spotlight_

### Official Review · Reviewer_d6S4 · 2023-10-31

**Soundness:** 3 good
**Presentation:** 2 fair
**Contribution:** 3 good
**Rating:** 8
**Confidence:** 2

**Summary:**

Masked image modeling methods (MIM) have started to employ discrete tokenization as a reconstruction target instead of raw pixel values. Yet, the role of tokenization and how it affects downstream performance is not well studied. This paper studies the impact of tokenizer design on downstream performance. The paper finds that that tokenizer that are more aligned with the downstream task labels (ie, more discriminative with respect to downstream classes) result in better performance, and design a metric to measure this similarity. The paper also introduces a new tokenizer based on clustering features to provide discretized reconstruction targets. Methods trained with the new tokenization achieve a better performance that existing approaches. The paper also includes some interesting analysis for the new tokenizer.

**Update (11/20):** I have raised my rating from 6 to 8 in respond to reading the other reviews and the author's response to all reviews. I think the paper presents an interesting study of the impact of discretization and proposes an interesting approach to representation learning.

**Strengths:**

- The framing of the equivalency structure induced by the tokenization was quite interesting.

- The paper poses an interesting question regarding the impact of tokenization and provides several interesting observations and analysis to answer it.

- The proposed metric is intersting and seems to have predictive power. Although, as noted in the weaknesses, it would be great to report it for all experiments.

- The proposed tokenizer is fairly simple, especially for K-MIM pixel which achieves a significant performance gain without requiring any other models.

**Weaknesses:**

- I found the name "Mixture Tokenizer" a bit confusing since it almost seems adversarial in nature, and wasn't sure if this naming is common in another sub-field. It seems to me that this would be an adversarial/worst-case tokenizer; one that implicitly learns a class structure that is "orthogonal" to the desired one (eg, one that learns backgrounds instead of foreground objects in a perfectly balanced datasets). If so, I would suggest naming it something that denotes this quality better simply a "mixture tokenizier."

- The use of a pre-trained SSL models to create the tokenization seems a bit odd. I would argue that if you use the features of model X to generate the labels/tokens to train model Y, then model Y is effectively being supervised by model X. While this is okay since both models have similar training requirements, one would expect model Y to outperform model X for this strategy to be pragmatic. Yet, K-MIM DINO achieves a much lower linear probe accuracy than DINO. Furthermore, the efficiency argument made in Sec 5.2 would need to take into account the time taken to train DINO for the K-MIM DINO results (the argument for K-MIM PIXEL holds and is a very nice finding).

- The introduction goes directly into specific methods that might not be known to the readers. Providing some smoother transition (eg, explaining what those methosd are doing first) could improve the readability of the introduction.

- The TCAS metric is only shown in Table 2, I think it would be nice to include it as a column in Tables 3 and 4 to see how well it explains variation in performance. Particularly, I am curious how much it changes with the choice of K in table 4, and whether that would explain some of the patterns there.

**Questions:**

- Could you explain why the bounds in equation 7 do not depend on $l$? Is it because each point is equivalent to an equal number of each class?

- He et al (CVPR 2022) reported a fine-tuning performance of 83.6, yet Table 3 reports 82.9. Could you please comment on this discrepancy?

- Table 3 notes that K-MIM DINO achieves a linear probe accuracy of 67.4, which is significantly lower than 78.2 reported by Caron et al (ICCV 2021), while outperforming them on fine-tuning (83.8 vs. 82.8). I was curious why you think the model underperforms this much despite being given being trained using the equivalency structure learned by DINO.

- The results reported in Table 4 are quite interesting as they indicate that performance deteriorates quickly for larger token books, while a larger number of tokens seems to benefit DINO. Could you please comment on this result? I would be curious if TCAS could shed some light on this.

- I think the finding that discretized features (HOG/DINO) provide better targets than raw pixels. DINO is an interesting target as it has been shown to be good local descriptor. I am curious if other features would be useful as well; eg, imagenet-supervised VIT features or MAE features. The first should be more aligned with the downstream task structure, while the latter is based on a model that seems to perform worse. I am curious if you had tried this or what your thoughts are on other features as targets for clustering.

---

> ### Author Response · Authors · 2023-11-20
> **Reply to Reviewer d6S4 (1/2)**
>
> We thank Reviewer d6S4 for the comments and the appreciation on our work. We address your concerns in the following points:
>
> ---
>
> Q1. The name "Mixture Tokenizer" is a bit confusing.  It seems to me that this would be an adversarial/worst-case tokenizer; one that implicitly learns a class structure that is "orthogonal" to the desired one (eg, one that learns backgrounds instead of foreground objects in a perfectly balanced datasets). If so, I would suggest naming it something that denotes this quality better simply a "mixture tokenizier."
>
> A1. Thanks for your thoughtful suggestion! We rename "Mixture Tokenizer" with "Cross-class Tokenizer", which reflects the essence of the tokenizer, emphasizing that each equivalence class comprises elements from multiple classes.
>
> ---
>
> Q2. The use of a pre-trained SSL models to create the tokenization seems a bit odd. K-MIM DINO achieves a much lower linear probe accuracy than DINO. Furthermore, the efficiency argument made in Sec 5.2 would need to take into account the time taken to train DINO for the K-MIM DINO results.
>
> A2. This setting mainly follows previous tokenization-based MIM works that adopt DINO as a supervision for MIM (like MaskFeat [1] and dBoT [2]). Nevertheless, as you pointed out, this setting is indeed a bit odd since DINO is already good enough, and it also introduces heavy pretraining cost when taken into account (in practice we just load public checkpoints).
>
> To resolve this issue, we further propose K-MIM PIXEL, which directly performs tokenization from raw pixels and thus have little pretraining cost. From Table 3, we can observe that K-MIM PIXEL significantly outperform those without pretrained models (e.g., raw pixels, HOG), and it can even outperform DINO supervision on finetuning (e.g., 86.4 vs 84.7 on ImageNet-100). Thus, we believe that K-MIM PIXEL is more practical solution for tokenization-based MIM training.
>
> [1] Wei et al. Masked Feature Prediction for Self-Supervised Visual Pre-Training, CVPR 2022.
>
> [2] Liu et al. Exploring Target Representations for Masked Autoencoders, Arxiv 2209.03917.
>
> ---
>
> Q3. The introduction goes directly into specific methods that might not be known to the readers. Providing some smoother transition (eg, explaining what those methosd are doing first) could improve the readability of the introduction.
>
> A3. Thanks for your thoughtful suggestion! We add more explanations into the introduction part in the revision.
>
> ---
>
> Q4. The TCAS metric is only shown in Table 2, I think it would be nice to include it as a column in Tables 3 and 4 to see how well it explains variation in performance. Particularly, I am curious how much it changes with the choice of K in table 4, and whether that would explain some of the patterns there.
>
> A4. Thanks for your suggestions! Since Table 3 also contains several methods without using discrete tokenization, we do not include TCAS in Table 3. Following your suggestions, we organize the TCAS score and performance of all the tokenizers in Appendix E in the revision. It can be indicated from Figure 4 that there is a strong linear correlation between TCAS and Linear Probing Accuracy in general. This indicates that TCAS does offer a guidance on assessing the performance of a tokenizer.
>
> ---
>
> Q5. Could you explain why the bounds in equation 7 do not depend on $l$? Is it because each point is equivalent to an equal number of each class?
>
> A5. No. It is because each class exhibit identical ratios of elements from the three sets $P_1/P_2$, $P_2/P_1$ and $P_1\cap P_2$. In fact, we can prove that if two equivalence classes $S_i$ and $S_j$ exhibit identical ratios of elements from the three sets $P_1/P_2$, $P_2/P_1$ and $P_1\cap P_2$, i.e. $n_{i,1}/n_{j,1}=n_{i,2}/n_{j,2}=n_{i,3}/n_{j,3}$, then the bound remains unchanged if we merge $S_i$ and $S_j$ into a new equivalence class. Since the proportions of elements from any of the three sets remain consistent across equivalence classes resulting from cross-class tokenization, the $l$ equivalence classes can merge into one without affecting the bound. Consequently, this explains why the bounds in Equation (7) exhibit no dependency on $l$.
>
> ---

---

> ### Author Response · Authors · 2023-11-20
> **Reply to Reviewer d6S4 (2/2)**
>
> Q6. [3] reported a fine-tuning performance of 83.6, yet Table 3 reports 82.9. Could you please comment on this discrepancy?
>
> A6. In [3], the fine-tuning performance of 83.6 is based on the checkpoints of 1600 epochs pretraining, where our setting is 200 epochs pretraining. Our 200-epoch result is consistent with the result in [4].
>
> [3] He et al. Masked Autoencoders Are Scalable Vision Learners, CVPR 2022.
>
> [4] Zhange et al. How Mask Matters: Towards Theoretical Understandings of Masked Autoencoders, NeurIPS 2022.
>
> ---
>
> Q7. Table 3 notes that K-MIM DINO achieves a linear probe accuracy of 67.4, which is significantly lower than 78.2 reported by Caron et al (ICCV 2021), while outperforming them on fine-tuning (83.8 vs. 82.8). I was curious why you think the model underperforms this much despite being given being trained using the equivalency structure learned by DINO.
>
> A7. Indeed, MIM methods will underperform contrastive methods in the linear probing accuracy and outperform them in the finetuning accuracy. There has been a previous work studying this phenomenon [5]. The following statements are quoted from [5]:
>
> >Contrastive learning methods outperform MIM in linear probing and small model regimes Contrastive learning will capture the shapes of the images, which will  help recognize objects and distinguish images. In contrast, MIM preserves texture and the diversity of representation, which may not correlate as strongly with objects or content when compared to the emphasis on shapes.
>
> Therefore, the phenomenon that K-MIM DINO will underperform DINO in the linear probing accuracy is due to the different training dynamics (MIM v.s. Contrastive Learning).
>
> [5] Park et al. What Do Self-Supervised Vision Transformers Learn? ICLR 2023.
>
> ---
>
> Q8. The results reported in Table 4 are quite interesting as they indicate that performance deteriorates quickly for larger token books, while a larger number of tokens seems to benefit DINO. Could you please comment on this result? I would be curious if TCAS could shed some light on this.
>
> A8. Indeed, the results reported in Table 4 indicate that the optimal clustering number for K-MIM DINO will be larger than K-MIM PIXEL. arises from the distinct separability of Kmeans performing in the pixel space and K-means performing in the DINO feature space [6]. To be specific, the dataset cannot be clustered well into real classes using K-means in the pixel space. Therefore, when increasing the number of clusters in K-means, it leads to the model overfitting noise or local features in the data, resulting in low-quality clusters and larger labeling error. Hence, the optimal clustering number for K-MIM is smaller. In the contrary, each class can be seperated well in the DINO feature space [7]. Therefore, a larger clustering number $K$ is optimal for the K-MIM DINO.
>
> [6] Caron et al. Deep Clustering for Unsupervised Learning of Visual Features
>
> [7] Caron et al. Emerging Properties in Self-Supervised Vision Transformers, ICLR 2023.
>
> ---
>
> Q9. I am curious if other features would be useful as well; eg, imagenet-supervised VIT features or MAE features. The first should be more aligned with the downstream task structure, while the latter is based on a model that seems to perform worse.
>
> A9. As you suggest, we use DeiT features and MAE features in our K-MIM methods, where DEiT is an supervised ViT method trained on ImageNet. We pretrain DeiT and MAE models using ViT-S on ImageNet-100 for 800 epochs. The evaluation setting is consistent with that in our paper. Since DeiT is more aligned with the downstream task structure and MAE performs worse, we expect K-MIM DeiT will have a smaller TCAS socre and achieve better results than K-MIM DINO. Conversely, we anticipate that K-MIM MAE will have a larger TCAS score and underperform K-MIM DINO. The results are shown below:
>
> |Method|TCAS score|Linear Probing Acc.|
> |---|---|---|
> |K-MIM DINO|0.15|59.7|84.7|
> |K-MIM DeiT|0.09|64.8|83.1|
> |K-MIM MAE|0.34|52.3|83.2|
>
> As the table indicates, the TCAS score aligns well with the linear probing accuracy. K-MIM DeiT with the smallest TCAS score achieves the best linear probing accuracy and K-MIM MAE with the largest TCAS score underperforms K-MIM DINO. This conclusion ensures the reliability of our TCAS metric. For more detailed information, please refer to Appendix E, where we organize the TCAS scores and performances of various tokenizers.
>
> ---
>
> Thanks for your comments and hope our answers could address your concerns. Please let us know if you have additional questions.

---

> > ### Comment · Reviewer_d6S4 · 2023-11-21
> > **Reviewer Response**
> >
> > Thank you for the clarification. I appreciate your thoroughness and willingness to include additional results. I think the paper is strong with the additional results and hope the other reviewers will agree as well. After reading the rebuttal and the other reviews, I will be raising my score as my concerns have been addressed.
> >
> > One last thing question that I had for the authors. The current paper presents a very interesting study on the impact of tokenization. One alternative is to use an online tokenizer as done by iBOT and scaled up by DINOv2. This seems to results in a very strong performance. Meanwhile, K-MIM with DeIT outperforms that other MIM methods, but still seems to be short of iBOT's performance despite being trained with the equivalency structure imposed by the ground-truth labels that has a very low TCAS score. I was curious what you thought about this, specifically:
> > - is comparing K-MIM DeiT with iBOT numbers a fair comparison to K-MIM DeiT or is there something that iBOT is leveraging that could balance the numbers?
> > - if an online tokenizer does indeed outperform a discrete tokenization using a groundtruth trained model's features, what should we infer from that? is this an issue due to the loss, the discretization, or something else?
> >
> > Thanks again for the clarification and I would appreciate your thoughts on the questions above.

---

> > > ### Author Response · Authors · 2023-11-22
> > > **Further Response to Reviewer d6S4**
> > >
> > > We thank Reviewer d6S4 for appreciating our response and highlighting its value to the field.  We will elaborate our explanations to answer your questions.
> > >
> > > ---
> > >
> > > Q1. K-MIM with DeIT outperforms that other MIM methods, but still seems to be short of iBOT's performance despite being trained with the equivalency structure imposed by the ground-truth labels that has a very low TCAS score. Is comparing K-MIM DeiT with iBOT numbers a fair comparison to K-MIM DeiT or is there something that iBOT is leveraging that could balance the numbers?
> > >
> > > A1. Indeed, it is not fair to directly compare K-MIM with iBOT, since the latter is not a pure MIM method. Specifically, iBOT also leverages a contrastive learning objective to align positive views (similar to DINO), which are known to deliver much superior performance than MIM on linear probing [1]. In this paper, we focus on inverstigating the MIM method itself, and do not incoporate contrastive learning objectives.
> > >
> > > [1] Liu et al. Exploring Target Representations for Masked Autoencoders, Arxiv 2209.03917.
> > >
> > > ---
> > >
> > > Q2. If an online tokenizer does indeed outperform a discrete tokenization using a groundtruth trained model's features, what should we infer from that? Is this an issue due to the loss, the discretization, or something else?
> > >
> > > A2. As discussed in A1, we believe that the superior performance of iBOT mainly comes from the contrastive learning part that it uses. It may also be regarded as an advantage of the online tokenizer, since it can be naturally integrated with other learning objectives.
> > >
> > > ---
> > >
> > > Thanks again for your detailed comments and encouraging score. Hope our explanations could ease your concerns. Please let us know if there is more to clarify.

---

> > > > ### Comment · Reviewer_d6S4 · 2023-11-22
> > > >
> > > > Thanks for the clarification. It seems the main explanation is the additional image-level contrastive loss. One final suggestion is to add this to the discussion or to even consider an iBOT baselined without the CLS token. Table 9 in iBOT [1] suggests that it does very poorly, although, it is unclear if this is due to them still using the CLS of the model which would receive no training in the final layer, or if this was accounted for somehow. Regardless, this is just a suggestion and I already recommend the paper for acceptance as mentioned in my previous comment. Thanks again for the engagement.
> > > >
> > > > [1] https://arxiv.org/pdf/2111.07832.pdf

---

### Official Review · Reviewer_f8wZ · 2023-10-31

**Soundness:** 3 good
**Presentation:** 3 good
**Contribution:** 3 good
**Rating:** 6
**Confidence:** 3

**Summary:**

This authors of this paper present theoretical analysis on the problem of masked image modeling (MIM), and study the impact of discrete tokenization in the MIM pipeline.

**Strengths:**

The authors present an intriguing tokenization approach using graph representation, and the paper is both technically robust and clearly articulated

**Weaknesses:**

The theoretical analysis is only considered for two classes. I wonder if this can be extended into multiple classes.

**Questions:**

Other the generalization to multi-classes. I have another question regarding the downstream error bound, it would be helpful if you can talk more in detail about the derivation perhaps in the appendix.

---

> ### Author Response · Authors · 2023-11-20
> **Reply to Reviewer f8wZ**
>
> We thank Reviewer f8wZ for the comments and the appreciation on our work. We address your concerns in the following points:
>
> ---
>
> Q1. The theoretical analysis is only considered for two classes. I wonder if this can be extended into multiple classes.
>
> A1. Sure. We add some discussions on the case of multiple classes in Appendix C. Similarly, we consider the three tokenization settings in this case and discuss how the tokenization influences the intra-class and inter-class connectivity, as well as the downstream error bound. In this situation, we still find that class-wise tokenization will have larger intra-class connectivity and cross-class tokenization will have larger inter-class connectivity. Therefore, similar to the two-class case in the paper, class-wise tokenization will enjoy a lower downstream error bound and cross-class tokenization will have a larger downstream error bound.
>
> ---
>
> Q2. I have another question regarding the downstream error bound, it would be helpful if you can talk more in detail about the derivation perhaps in the appendix.
>
> A2. We provide more detailed derivation process regarding the calculation of downstream error bound in Appendix B.
>
> ---
>
> Thanks for your comments and hope our answers could address your concerns. Please let us know if you have additional questions.

---

### Official Review · Reviewer_ss62 · 2023-11-01

**Soundness:** 4 excellent
**Presentation:** 3 good
**Contribution:** 3 good
**Rating:** 6
**Confidence:** 4

**Summary:**

The paper analyzes the effect of discrete tokens as targets of masked image model (MIM) training. Using a graph-based view for MIM [A], the paper shows that discrete tokens change the graph of MIM. Also, it provides a theorem that discrete tokens similar to image class are the best for MIM targets. Based on the theorem, a metric to measure better tokenizer for MIM is proposed, named Token-Class Alignment Similarity (TCAS). Last, it is shown that simple K-means, which have a better TCAS score than other baselines, could improve the performance of MIM.

[A] How Mask Matters: Towards Theoretical Understandings of Masked Autoencoders, NeurIPS 2022

**Strengths:**

- Theoretical analysis on a discrete tokenization method looks novel and interesting.
- A metric for tokenization (TCAS) would help a lot of researchers to investigate MIM.

**Weaknesses:**

- I think using discrete tokenization is not a mainstream of MIM. Representative methods, such as MAE, MaskFeat, and data2vec, demonstrate impressive performance without the discrete tokens. Thus, the contribution of the paper is hard to cover diverse variants of MIM.

- According to theorem 1, image classification training could be the best way to downstream error bound. But, in practice, MIM works better than classification training in a lot of cases. Thus, I doubt the general applicability of this theorem and the metric (TCAS) on diverse MIM tasks.

- Experimental results are limited to a small dataset (ImageNet-100) and short training (200 epochs on IN-1k). I think it is not enough to validate the effect of K-MIM. Reported numbers on TCAS are also limited.

**Questions:**

- Using DINO as a target representation is similar to [B]. Is there any relation between the distillation target and the TCAS metric? I think the paper would be better to cite [B] and add a discussion on it.

[B] Exploring Target Representations for Masked Autoencoders, arxiv

- According to theorem 1, it looks like an image classifier would be the best tokenizer for MIM. What is the TCAS score and MIM performance when using an image classifier, such as DeiT, as a tokenizer?

- Section 3 is explained with tokenization for a group-of-tokens, i.e. $x_2 \in R^{n \times s}$. But, in Section 4, it seems the tokenization is conducted for a single token. Is it possible to generalize a theorem from the group-of-tokens case to the single-token scenario?

- Discrete tokenizers like dVAE and VQGAN employ ConvNet or ViT, utilizing the entire image to create tokens. These tokens are interrelated, and a token from one location can incorporate patches from others. However, it looks like the paper handles these tokens as individual local information, which is not correct. Is there any explanation for this?

- Although K-MIM has better TCAS than other tokenizers, it is hard to say that K-MIM is specially designed for TCAS. Is there any discrete tokenization method optimized for the TCAS metric?

---

> ### Author Response · Authors · 2023-11-20
> **Reply to Reviewer ss62 (1/3)**
>
> We thank Reviewer ss62 for careful reading and detailed comments. We address your concerns in the following points:
>
> ---
>
> Q1. I think using discrete tokenization is not a mainstream of MIM. Representative methods, such as MAE, MaskFeat, and data2vec, demonstrate impressive performance without the discrete tokens.
>
> A1. Although discrete tokenization is not a default choice in MIM, it has been receiving wider and wider attention recently. In fact, one of the first MIM papers BeiT [1] uses tokenization, and so does iBoT [2]. Compared to those relying on raw inputs, the tokenization-based MIMs generally achieve much higher linear probing accuracy, which is known to benefit zero-shot generalization.
>
> Very recently, there is a phenomenological work [3] showing that tokenization is the key to visual generation. With the help of a good tokenizer, they attain the **best generation quality on ImageNet** simply with language model, surpassing previous sophiscated diffusion models. Similarly, visual tokens are also widely adopted by recent developments of visual foundation models. For example, equipped with visual tokenizers, MAGE [4] achieves **SOTA  generation ability within MIM methods**, and BEiT-3 [5] demonstrates **SOTA transfer performance on both vision-only and vision-language tasks**.
>
> Therefore, the importance of tokenization is increasingly recognized in visual tasks. And our investigation from a theoretical perspective provides timely insights into the mechanism behind it. As the first theoretical work in this direction, we believe that it could help inspire more researchers to investigate the use of tokenization for visual representation learning.
>
> [1] Bao et al. BEiT: BERT Pre-Training of Image Transformers, Arxiv 2106.08254.
>
> [2] Zhou et al. Image BERT Pre-training with Online Tokenizer, ICLR 2022.
>
> [3] Yu et al. Language Model Beats Diffusion -- Tokenizer is Key to Visual Generation, Arxiv 2310.05737.
>
> [4] Li et al. MAGE: MAsked Generative Encoder to Unify Representation Learning and Image Synthesis, CVPR 2023.
>
> [5] Wang et al. BEiT Pretraining for All Vision and Vision-Language Tasks, CVPR 2023.
>
>
> ---
>
> Q2. According to Theorem 1, image classification training could be the best way to downstream error bound. But, in practice, MIM works better than classification training in a lot of cases. Thus, I doubt the general applicability of this theorem and the metric (TCAS) on diverse MIM tasks.
>
> A2. We note that in existing literature, MIM has many downstream tasks, e.g., classification, detection, segmentation, etc. Among classification, there are two settings: linear probing (frozen features) and finetuning (tuning all features). Theorem 1 is established for the linear probing setting, which is the common evaluation setting in SSL theory, e.g., [6,7]. And indeed, the empirical results align well with our theory, since **MAE pretrained features** (no finetuning) underperform supervised learning by a large margin in this setting, as shown below.
>
> |Method|Epochs|Accuracy (%)|
> |---|---|---|
> |Supervised|300|77.9|
> |MAE|200 pretrain + 100 linear probing|55.4|
>
> **Difficulty in theoretical analysis.** Meanwhile, as you pointed out, MIM features often perform better on other settings when finetuning is allowed. However, we are not aware of any existing theory on these tasks. Finetuning is hard for theoretical analysis since the MAE pretraining weights only serve as the initialization, and SGD training on NNs is highly non-contex.
>
> **Correlation between tasks.** Nevertheless, we note that the downstream performance of MIM on other datasets is often correlated well with its linear classification performance [8]. Therefore, although Theorem 1 is based on linear classification, we believe the theorem still provides valuable guidelines for improving its performance on other MIM tasks.
>
> [6] Zhang et al. How Mask Matters: Towards Theoretical Understandings of Masked Autoencoders, NeurIPS 2022.
>
> [7] Cao et al. How to understand masked autoencoders, Arxiv 2202.03670.
>
> [8] Li et al. Benchmarking Detection Transfer Learning with Vision Transformers, Arxiv 2111.11429.

---

> ### Author Response · Authors · 2023-11-20
> **Reply to Reviewer ss62 (2/3)**
>
> Q3. Experimental results are limited to a small dataset (ImageNet-100) and short training (200 epochs on IN-1k). I think it is not enough to validate the effect of K-MIM. Reported numbers on TCAS are also limited.
>
> A3. Thanks for your suggestions! We have added additional experiments involving 800 epochs of pretraining on ImageNet-1k using ViT-B as the backbone. The comparisons with baseline methods are presented in the table below:
>
> |Method|Epochs|Linear Acc.|Finetuning Acc.|
> |---|---|---|---|
> |MAE|1600|68.0|83.6|
> |BEiT|800|N/A|83.2|
> |K-MIM PIXEL|800|69.4|83.6|
>
> Notably, with only half the number of pretraining epochs, K-MIM PIXEL reaches the same finetuning accuracy with MAE and even ourperforms MAE in the linear probing accuracy. The results indicate that K-MIM PIXEL outperforms the baseline results in the classification tasks, ensuring the better learning ability of K-MIM. We also add this experiments in Appendix D in the revision.
>
>
> ---
>
> Q4. Using DINO as a target representation is similar to [9]. Is there any relation between the distillation target and the TCAS metric? I think the paper would be better to cite [9] and add a discussion on it.
>
> A4. Our proposed method, MaskFeat, and dBot [9] can all leverage representations from a pretrained model like DINO as the prediction target, with variations in their processing methods. The distinctions are outlined in the following table:
>
> |Method|Process Method|
> |---|---|
> |MaskFeat|Directly use the representations|
> |dBot|Initially employs the representations, updating them per certain epochs|
> |K-MIM|Use the K-means results of the representations|
>
> Since the prediction target of dBot remains continuous, and its training is conducted in a multi-stage manner, our TCAS metric is designed specifically for discrete targets. As such, we will leave the analysis of distillation target of dBot to future works. We have cited [9] and incorporated the above discussion into Section 2 in the revision.
>
> [9] Liu et al. Exploring Target Representations for Masked Autoencoders, Arxiv 2209.03917.
>
> ---
>
> Q5. According to theorem 1, it looks like an image classifier would be the best tokenizer for MIM. What is the TCAS score and MIM performance when using an image classifier, such as DeiT, as a tokenizer?
>
> A5. We list the comparison of different tokenizers on ImageNet-100 using ViT-S with 200 epochs training in the table below for better illustration. The clustering number is set to $K=100$.
>
> |Method|TCAS score|Linear Probing Acc.|
> |---|---|---|
> |K-MIM PIXEL|0.52|50.1|
> |K-MIM DINO|0.15|59.7|84.7|
> |K-MIM MAE|0.34|52.3|83.2|
> |K-MIM DeiT|0.09|64.8|83.1|
>
> The results indicate that employing DeiT as a tokenizer obtains the lowest TCAS score and outperforms other tokenizers in the linear probing accuracy, which aligns well with Theorem 1. For more detailed information, please refer to Appendix E, where we organize the TCAS scores and performances of various tokenizers.
>
> ---

---

> ### Author Response · Authors · 2023-11-20
> **Reply to Reviewer ss62 (3/3)**
>
> ---
>
> Q6. Section 3 is explained with tokenization for a group-of-tokens, i.e. $x_2\in R^{n\times s}$. But, in Section 4, it seems the tokenization is conducted for a single token. Is it possible to generalize a theorem from the group-of-tokens case to the single-token scenario?
>
> A6. Indeed, in Section 3, our theory deals with general tokenizers that involve interactions among patches. However, in practice, it leads to a **exponential large space**, since each patch representation can be unique. However, when designing a practical metric in Sec 4, it is intractable for us to deal with. So we consider a bag-of-word model [10] for patch representations during the analysis in Sec 4, making it computationally tractable. Since patch representations are often local, this simplification (as a 1-gram model) also makes sense. In practice, we find the derived TCAS metric has good alignment with downstream performance, indicating that it could provide insights for the general cases. We have added clear explanations on these relationship in the beginning of Section 4.1.
>
> [10] Zellig Harris. Distributional Structure, doi:10.1080/00437956.1954.11659520
>
> ---
>
> Q7. Discrete tokenizers like dVAE and VQGAN employ ConvNet or ViT, utilizing the entire image to create tokens. These tokens are interrelated, and a token from one location can incorporate patches from others. However, it looks like the paper handles these tokens as individual local information, which is not correct. Is there any explanation for this?
>
> A7. Please see A6.
>
>
> ---
>
> Q8. Although K-MIM has better TCAS than other tokenizers, it is hard to say that K-MIM is specially designed for TCAS. Is there any discrete tokenization method optimized for the TCAS metric?
>
> A8. The guiding principle for TCAS is to assess the dissimilarity between the discrete token class and the true label. **Hence, the optimal choice for the TCAS metric is simply utilizing the true label as the tokenization.** In this context, our aim in the paper is to explore how to optimize the TCAS metric **when label information is unavailable**. Employing clustering methods such as K-means is likely to group samples with the same label together. Therefore, K-MIM is designed to optimize TCAS **under the condition of lacking label information.**
>
> ---
>
> Thanks for your comments and hope our answers could address your concerns. Please let us know if you have additional questions.

---

> ### Comment · Reviewer_ss62 · 2023-11-22
>
> Thank you for your response.
>
> Your response has addressed all of my questions.
> I'm satisfied with the revision.
>
> I think short responses can not solve the major weaknesses.
> - I knew that some papers use discrete tokens. But, still, discrete tokenization isn't an essential part of MIM, the paper can only contribute a specific range of MIM.
> - According to A2, A3, and A8, the contribution is focused on linear probing rather than fine-tuning of MIM. Because I think fine-tuning performance is the most surprising factor of MIM, it is a major weakness of the paper.
>
> But, I think the paper has a limited yet original contribution, and I slightly lean toward acceptance.
> I adjusted my rating to marginal acceptance.

---

> > ### Author Response · Authors · 2023-11-22
> > **Furthur Response to Reviewer ss62**
> >
> > We thank Reviewer ss62 for appreciating our response and the originality of our contribution. We will elaborate our explanations to address your concerns on the weaknesses.
> >
> > ---
> >
> > Q1. I knew that some papers use discrete tokens. But, still, discrete tokenization isn't an essential part of MIM, the paper can only contribute a specific range of MIM.
> >
> > A1. Indeed, we agree that tokenization is not necessary for MIM. In fact, **tokenization and MIM are two overlapped areas.** Discrete tokenization also finds wide applications in improving image generation [1], tranferability [2], and robustness [3], never to say, model compression [4] and so on. Our paper firstly analyzes the role of discrete tokenization from a representation learning perspective, explain the source of its benefits, and propose a metric to evaluate it. In this paper, for a concrete discussion, we take MIM as an example, but we believe that the analysis is also helpful for understanding its effectiveness on the other areas as well. So, we believe that the significance of our work is not limited to a specific class of MIM methods, but can inspire researches in broader areas as well.
> >
> > [1] Yu et al. Language Model Beats Diffusion -- Tokenizer is Key to Visual Generation, arXiv:2310.05737.
> >
> > [2] Jin et al. Unified Language-Vision Pretraining in LLM with Dynamic Discrete Visual Tokenization, arXiv:2309.04669.
> >
> > [3] Mao et al. Discrete Representations Strengthen Vision Transformer Robustness, ICLR 2022.
> >
> > [4] Fu et al. Contrastive Quant: Quantization Makes Stronger Contrastive Learning, ACM DAC 2022.
> >
> > ---
> >
> > Q2. According to A2, A3, and A8, the contribution is focused on linear probing rather than fine-tuning of MIM. Because I think fine-tuning performance is the most surprising factor of MIM, it is a major weakness of the paper.
> >
> > A2. Indeed, fine-tuning performance is the most surprising factor of MIM. However, finetuning poses challenges for theoretical analysis since the pretraining weights only serve as the initialization, and SGD training on NNs are highly non-contex. To our knowledge, there does not exist any theory on these tasks. Additionally, we observe positive correlation between the linear probing performance and the finetuning performance within MIM methods. For example, as shown in Table 7 in the MAE paper [5], the linear probing accuracy synchronously grows with the finetuning accuracy as the training epochs increase. Our proposed K-MIM PIXEL also achieves significantly better finetuning accuracy than MAE (K-MIM PIXEL 86.4% v.s. MAE 81.8% on ImageNet100), despite K-MIM being designed under the guidance of linear probing-based theory. Therefore, we believe that contribution of our work can extend beyond linear probing evaluation; it may provide guidance for MIM across broader evaluation metrics as well.
> >
> > [5] He et al. Masked Autoencoders Are Scalable Vision Learners, CVPR 2022.
> >
> > ---
> >
> > Thanks again for your detailed comments and appreciation on the originality of our work. Hope our explanations could ease your concerns. Please let us know if there is more to clarify.

---

### Official Review · Reviewer_yDgo · 2023-11-01

**Soundness:** 4 excellent
**Presentation:** 4 excellent
**Contribution:** 3 good
**Rating:** 8
**Confidence:** 3

**Summary:**

This paper explores the role of tokenization in Masked image modeling (MIM). It starts by discussing how discrete tokenization affects generalization performance on downstream tasks. For measuring the proficiency of different tokenization roles, this work designs the token-class alignment similarity (TCAS) metric. Based on the TCAS, they propose a MIM framework (K-MIM) with a cluster-based discrete tokenization scheme. Extensive experiments are conducted to validate the effectiveness of the proposed tokenization strategy.

**Strengths:**

- The main problem, "What is the role of tokenization in MIM? How does it affect downstream performance?", is interesting and necessary.
- The proposed token-class alignment similarity (TCAS) is a cheap but effective metric to measure the performance of tokenization roles.
- This paper theoretically and empirically demonstrates the superiority of class-wise tokenization.
- The paper is well-organized and easy to follow.

**Weaknesses:**

- It would be better to show more ablation results about clustering numbers in Table 4.

**Questions:**

- Could you please show more ablation results about clustering numbers (e.g., K = 10, 25, 200) in Table 4?

---

> ### Author Response · Authors · 2023-11-20
> **Reply to Reviewer yDgo**
>
> We thank Reviewer yDgo for the appreciation on our work. We address your concern in the following point:
>
> ---
>
> Q1. Could you please show more ablation results about clustering numbers (e.g., K = 10, 25, 200) in Table 4?
>
> A1. Sure. We conduct K-MIM PIXEL experiments on ImageNet-100 using ViT-Small, with clustering numbers $K=10, 25, 200$. The results (along with $K=50,100,1000$) are shown below.
>
> |Clustering Numbers $K$| LP Acc | FT Acc|
> |---|---|---|
> |10|42.7|81.9|
> |25|48.3|84.6|
> |50|52.7|86.4|
> |100|50.1|83.7|
> |200|50.8|84.8|
> |1000|50.6|84.4|
>
> The results indicate that the performance peaks at $K=50$ and gradually declines as $K$ decreases from this optimal value. The diminished performance for small $K$ is due to the large inter-class connectivity and large labeling errors, where we have discussed in Section 3 that these factors lead to a larger error bound and consequently result in poorer performance.
>
>
>
> ---
>
> Thanks for your comments and hope our answers could address your concerns. Please let us know if you have additional questions.

---

> > ### Comment · Reviewer_yDgo · 2023-11-22
> > **Reviewer Response**
> >
> > Thanks for the response, I have decided to maintain my original score.

---

### Meta-Review · Area_Chair_QypC · 2023-12-05

**Metareview:**

This paper discusses the role of discrete visual tokens in Masked Image Modeling (MIM), a recently popularized SSL technique. It introduces the Token-Class Alignment Similarity (TCAS) metric for evaluating the effectiveness of different tokenization methods and a novel MIM framework, K-MIM, employing a cluster-based discrete tokenization scheme. The authors empirically validated this approach across various benchmark datasets and highlighted the synergy between discrete tokenization and MIM's overall effectiveness.

The reviewers found the presented ideas and results useful. Given the discussions and the increasing popularity MIM approaches, I will recommend acceptance.

**Justification For Why Not Higher Score:**

- Discrete tokens are not yet mainstream in MIM.
- Theory covers only a small subset of potential use cases.

**Justification For Why Not Lower Score:**

- MIM is one of the most popular recent SSL approaches
- Potentially a large audience at ICLR.

---

### Decision · Program_Chairs · 2024-01-16

Accept (spotlight)